# Hippocampal Transcriptome Analysis in a Mouse Model of Chronic Unpredictable Stress Insomnia

**DOI:** 10.3390/biomedicines13051205

**Published:** 2025-05-15

**Authors:** Shuo Zhang, Changqing Tong, Na Cao, Dong Tian, Linshan Du, Ya Xu, Weiguang Wang, Zijie Chen, Shuangqing Zhai

**Affiliations:** 1Beijing University of Chinese Medicine Third Affiliated Hospital, Beijing 100013, China; shuoz357642@163.com; 2College of Traditional Chinese Medicine, Beijing University of Chinese Medicine, Beijing 100029, China; 18811358276@163.com (C.T.); 20240941003@bucm.edu.cn (N.C.); 20180931012@bucm.edu.cn (D.T.); 17602668073@163.com (L.D.); xuya@bucm.edu.cn (Y.X.); 202001031@bucm.edu.cn (W.W.)

**Keywords:** chronic unpredictable mild stress, insomnia model, transcriptome, albumin

## Abstract

**Background**: This study aimed to develop a model for understanding stress-induced sleep disturbances and to explore the potential interactions between sleep disturbances and mood disturbances. **Methods**: The chronic unpredictable mild stress (CUMS) group was established using the CUMS method, while the CUMS+Noise group was subjected to an additional 8-h exposure to noise in conjunction with the CUMS protocol. Each group was tested for anxiety and depressive-like behavior using the open-field, elevated plus maze, tail suspension, and forced swimming tests in male C57BL/6J mice. Subsequently, we assessed sleep status using sleep recordings and a standardized scoring system alongside the pentobarbital sodium-induced sleep test. **Results**: The mice in both model groups exhibited anxiety-like behavior. Sleep disturbances observed in the CUMS+Noise group were characterized by disruptions in sleep duration and circadian rhythm. This observation was supported by a marked reduction in multiple sleep time intervals and single sleep duration, as well as a significant increase in sleep duration at the final time interval of ZT23-24. To further investigate the potential mechanisms of interaction, we conducted an analysis of hub genes present in the hippocampal sequencing data utilizing weighted gene co-expression network analysis (WGCNA). Pearson correlation analysis revealed a significant association between the hub genes *Alb*, *P2rx1*, and *Npsr1* and key phenotypic traits. However, PCR experiments indicated that only *Alb* showed a significant difference, which aligns with the sequencing results. **Conclusions**: Albumin is a crucial transporter protein for thyroid hormones and plays a vital role in their metabolism. The interaction between sleep disorders and anxiety-like behavior may be closely linked to the dysfunctional transportation of thyroid hormones by albumin.

## 1. Introduction

As the incidence of sleep disorders and chronic stress increases in contemporary society, a growing body of research has substantiated the intricate associations between sleep and stress [1,2]. Numerous studies have indicated that prolonged exposure to chronic stress is associated with an elevated risk of developing insomnia. Failure to address or alleviate psychological or environmental stress in a timely manner may lead to the progression of stress-related insomnia to chronic insomnia [3,4]. Epidemiological studies have suggested that approximately 10% to 15% of the adult population experience chronic insomnia, while another 25% to 35% suffer from transient or occasional insomnia [5]. Chronic insomnia is a global public health issue, imposing substantial burdens on both families and society at large [6]. It is important to understand that chronic stress is linked not just to sleep problems but also to the development of mood disorders [7]. Consequently, anxiety and depressive disorders are often observed to co-occur with insomnia, frequently manifesting as comorbid or concomitant symptoms. Studies have shown that more than 60% of people diagnosed with generalized anxiety disorder (GAD) also suffer from co-occurring sleep disorders. Sleep disturbances are typically characterized by recurrent awakenings, reduced overall sleep duration, and decreased slow-wave sleep [8,9]. Symptoms associated with anxiety and depressive disorders, particularly persistent worry, can contribute to the onset of insomnia. Conversely, insomnia may exacerbate symptoms of anxiety and depression. Clinical treatments for anxiety and depression have been shown to alleviate insomnia-related symptoms [10,11]. This suggests the potential for shared mechanisms between stress-related insomnia and mood disorders. However, the specific interactions underlying these relationships still require further investigation.

The development of a reliable animal model for stress-induced insomnia represents a critical advancement for mechanistic investigations and therapeutic discovery. The current models used to study stress-induced insomnia include the post-traumatic stress disorder (PTSD) model [12], the chronic unpredictable mild stress (CUMS) combined with sleep deprivation (SD) model [13], the chronic restraint stress (CRS) model [14], the chronic social failure stress model [15], the rat dirty cage replacement stress model [16], and the permanent water avoidance stress on wheels model [17]. Notably, the combined CUMS/CRS model with sleep deprivation is considered to replicate clinical symptoms of stress-induced insomnia more accurately. Therefore, based on previous studies, it is crucial to perform further research on the preparation conditions of this stress-induced insomnia model and its biological mechanisms. Several studies carried out on rodent models have illustrated the effects of stress on both mood and sleep quality. For example, research has demonstrated that the CUMS model induces anxiety and depression-like behavior in mice, which show impairments in hippocampal neurotransmission and synaptic plasticity [18]. Similarly, the use of both the CUMS and SD models to induce insomnia led to notable reductions in sleep latency and sleep duration, along with impairments in the synaptic plasticity of the hippocampus [19]. Additionally, another study found that mice subjected to the CRS model experienced a notable increase in the length of their rapid eye movement (REM) sleep compared to the control group, regardless of whether the conditions were dark or illuminated [20]. Further supporting this, in a daily water immersion and three-week restraint stress model, mice exhibited increased duration of non-rapid eye movement (NREM) sleep time at low delta-band power densities [21]. In conclusion, the combination of CUMS/CRS with SD can be used to establish a stress-induced insomnia model.

There is a lack of comprehensive understanding in existing research regarding how stress-related mood changes relate to sleep disturbances. In this study, a mouse model of stress-induced insomnia was established by integrating chronic unpredictable stress and sleep deprivation, as outlined in earlier research [13]. We examined behavioral tests, sleep–wake cycle scoring, and electrophysiological recording data, along with an analysis of the hippocampal transcriptome in the model mice. This study aimed to establish a fitting animal model for insomnia triggered by stress. On this basis, underlying mechanisms of interaction between stress-related insomnia and anxiety-like behavior were explored. The results obtained provide valuable insights into the interactions between chronic stress-induced sleep disorders and mood disorders.

## 2. Materials and Methods

### 2.1. Animals

Specific pathogen-free (SPF) male C57BL/6J mice, aged 6–8 weeks and weighing 20 ± 2 g, were purchased from Beijing Vital River Laboratory Animal Technology Co., Ltd. (Beijing, China), with experimental animal license no. SYXK (Beijing) 2023-0011. The 27 mice were randomly divided into control, CUMS model, and CUMS+Noise model groups. They were maintained at the Institute of Animal Experiment Centre, Beijing University of Chinese Medicine. The mice had free access to food and water and were maintained at a temperature of 22.0–24.0 °C and a relative humidity of 60–70%. The experiments started after 7 days of acclimatization. The experiments were conducted in accordance with the National Institutes of Health Guide for the Care and Use of Laboratory Animals and were reviewed and approved by the Experimental Animal Ethics Committee of Beijing University of Chinese Medicine (no. BUCM-20240521102-2120). In this study, we established the zeitgeber time (ZT), with ZT0 defined as the time of light onset in the morning (8:00 a.m.) and ZT12 as the time of light offset in the evening (8:00 p.m.). Each day was divided into two phases: a light phase (ZT0-12) and a dark phase (ZT12-24) [22].

### 2.2. Instruments and Reagents

Open-field box, elevated plus maze, tail suspension box, and forced swim test apparatus (Beijing ZSdichuang Technology Development, Beijing, China); Ethovision XT 15 behavioral analysis software (Noldus, Wageningen, The Netherlands); 9700 ordinary PCR instrument (Applied Biosystems, Foster City, CA, USA); NanoDrop 2000 UV spectrophotometer (Thermo Fisher Scientific, Waltham, MA, USA).

### 2.3. CUMS and CUMS+Noise Model

We referenced the study conducted by Fang et al. [13]. A mouse model of chronic unpredictable stress-induced anxiety combined with insomnia was established according to their protocol. The model was developed over a duration of 28 days, and a timeline diagram is shown in Figure 1A. The mice in the model group were randomly exposed to the following 8 stressors during the light phase: ① gentle tail restraint for 3 min; ② restraint for 4 h; ③ swimming in cold water at 10 °C for 5 min; ④ swimming in warm water at 45 °C for 5 min; ⑤ shake cage horizontally for 10 min; ⑥ water fast for 12 h; ⑦ food fast for 12 h; ⑧ wet bedding for 12 h. In addition, the model mice received the following 6 stressors during the dark period (12 h): ① wet bedding; ② light (300 lux); ③ crowding; ④ tilting the cage at 45°; ⑤ water fasting; ⑥ food fasting. In addition, no more than 5 stressors were delivered during the light phase and no more than 6 stressors were delivered during the dark phase so that the mice could not anticipate the occurrence of the stressors. Furthermore, beginning on day 15, the CUMS+Noise model group was exposed to traffic noise that ranged from 70 to 80 dB, lasting 8 h daily for a total of 14 days [23]. The control group received free food and water.

After the behavioral tests, the mice were fasted for 12 h, and they were anaesthetized with pentobarbital sodium at ZT0 on the next day. Blood was collected via retro-orbital sampling under anesthesia. The blood samples were subjected to centrifugation at 3000 rpm for 15 min to obtain serum. The hippocampus of each mouse was collected using a freezing tube and immediately stored at −80 °C for subsequent investigations, such as qRT-PCR.

### 2.4. Pentobarbital Sodium-Induced Sleep Test (PSST)

The protocol was established by Ying-Jie Dong et al. [24]. The pentobarbital sodium-induced sleep test was performed on the last day of model preparation. The mice were administered an intraperitoneal injection of pentobarbital sodium at a dosage of 50 mg/kg, which was established as the hypnotic dose and represented the upper limit determined prior to the experiment. Sleep onset in the mice was operationally identified by the loss of the righting reflex lasting longer than one minute. The righting reflex is a commonly used parameter to assess sleep onset in rodent models. Each mouse’s timing related to the loss and recovery of the righting reflex was precisely noted, emphasizing the aspects of sleep latency (SL) and sleep duration (SD). SL was defined as the time between the injection of pentobarbital sodium and the loss of the righting reflex. SD was defined as the time interval between the loss and recovery of the righting reflex.

### 2.5. Behavioral Tests

#### 2.5.1. Open-Field Test (OFT)

The protocol was established by Hui-juan Hu et al. [25]. First, all of the mice were placed in the behavioral laboratory for 1 h. The open-field test was conducted in a 50 × 50 × 50 cm square box with the camera directly above the box, and the test was conducted during the light phase at ZT1-4. Each mouse was placed separately in the central arena of the test box (20 × 20 cm) and allowed to move freely for approximately 5 min while being recorded by an overhead camera. Subsequently, the mouse was removed from the open-field box, and the walls and bottoms of the container were cleaned using 75% alcohol. The other mice were subjected to the same experimental procedure. The footage was analyzed using Ethovision XT 15 software (Noldus, Wageningen, The Netherlands) for the following parameters: percentage of total distance moved in the center (TDMC%), time spent in the center (TPC), percentage of total distance moved in the periphery (TDMP%), and time spent in the periphery (TPP).

#### 2.5.2. Elevated Plus Maze Test (EPM)

The protocol was adapted from Hui-juan Hu et al. [25]. First, all of the mice were placed in the behavioral laboratory for 1 h. The elevated plus maze consisted of two closed arms (length 65 cm, width 5 cm, height 20 cm) and two open arms (length 65 cm, width 5 cm). At the center of the device was an open space measuring 5 × 5 cm. The maze was 50 cm above the ground. The test was conducted during the light phase at ZT1-4. At the beginning of the test, each mouse was individually placed in the central area, facing the open-arm area. The mouse was free to explore for 5 min while being recorded by an overhead camera. Subsequently, the mouse was removed from the maze, and the walls and bottoms of the maze were cleaned using 75% alcohol. The other mice were subjected to the same experimental procedure. The percentage of entries into the open arms (OAE%), the time spent (OAT%), and the total distance moved (OATDM%) were analyzed using Ethovision XT 15 software (Noldus, Wageningen, The Netherlands).

#### 2.5.3. Tail Suspension Test (TST)

The protocol for the tail suspension test (TST) was as previously described by Fu-Rong Xu et al. [26]. All of the mice were placed in the behavioral laboratory for 1 h. The test was conducted during the light phase at ZT1-4. Subsequently, the mice were suspended by the tail using adhesive tape placed 2 cm from the tip. Each mouse was placed in a suspension chamber that was fully isolated from sound and sight for 5 min. Then, the mouse was removed from the chamber, and the walls and bottoms of the chamber were cleaned using 75% alcohol. The other mice were subjected to the same experimental procedure. Immobility time was defined as passive suspension or complete absence of movement in the mice, which was analyzed using Ethovision XT 15 software (Noldus, Wageningen, The Netherlands). The system defined the duration by establishing a specific threshold tailored to each individual mouse. This threshold successfully distinguished between active movement and stillness, excluding any moments of activity and focusing solely on immobility time.

#### 2.5.4. Forced Swimming Test (FST)

The protocol for the forced swim test (FST) was as previously described by Fu-Rong Xu et al. [26]. The test was conducted during the light phase at ZT1-4. All of the mice were placed in the behavioral laboratory for 1 h. Subsequently, each mouse was individually placed in a Plexiglas cylinder (30 cm high × 15 cm diameter), which was filled with water (24 ± 1 °C) to a height of approximately 15 cm to prevent the mouse from reaching the bottom. The water was changed before each test. The mice were placed in water for 5 min and then returned to their cages. Immobility time was defined as the minimal movements necessary to keep the head above water or complete absence of movement in the mice, which was analyzed using Ethovision XT 15 software (Noldus, Wageningen, The Netherlands).

### 2.6. Sleep Recordings and Scoring

Data on sleep and wake activities were collected using a computerized piezoelectric system, namely, the PiezoSleep model developed by Signal Solutions, located in Lexington, Kentucky. Two piezoelectric films were placed underneath the cages of the subject mice, which were capable of detecting pressure changes, the intensity and duration of which were scored by a computerized scoring program (SleepStat; Signal Solutions, Lexington, KY, USA) [27]. An automatic score of <0 represents wake, while a score of >0 represents sleep. Piezoelectric systems have been validated to reach ~90% concordance with EEG/EMG-based sleep recordings, but EEG/EMG remains the gold standard [28]. Signals were extracted from short-duration pressure signal segments and automatically classified every 2 s during the 12 h dark phase. The collected data were binned at 5 min intervals using a rolling average of sleep percentages, and the average duration of wake or sleep episodes was calculated based on the individual arousal array duration. In addition, the timing of the duration of sleep episodes was only initiated when more than 50% of the 30 s interval was spent in wakefulness, and it was terminated when less than 50% of the 30 s interval was spent in wakefulness [29]. The detection time occurred during the 12 h dark phase at ZT12–24, corresponding to 20:00 to 8:00 in a standard cycle where ZT0 = lights on at 8:00 and ZT12 = lights off at 20:00.

### 2.7. RNA Sequencing

At least 1 μg of total RNA was isolated from the hippocampus (5 randomly selected samples per group) for RNA library preparation using an NEBNext^®^ UltraTM RNA Library Prep Kit^®^ (NEB, Ipswich, MA, USA). Messenger RNA was purified from total RNA using oligonucleotide (dT) magnetic beads, followed by heating to fragment the RNA. Random hexamer primers and reverse transcriptase were used to synthesize cDNA from RNA, and inhibitors of DNA contamination were included to prevent the use of DNA as a template. The estimated size of the final PCR product was between 250 and 300 bp. PCR amplification was then performed, and the PCR product was purified using AMPure XP beads to obtain the final library. The quality and quantity of the libraries were assessed using Bioanalyzer or qPCR, and subsequently, the libraries were pooled. The different libraries were pooled for Illumina sequencing according to the validated concentration and downstream target data requirements [30]. All sequencing raw data were uploaded to the NCBI database (submission ID: SUB14868366; BioProject ID: PRJNA1186268).

### 2.8. WGCNA Analysis and Key Module Bioinformatics Analysis

To identify gene modules associated with the phenotypic traits of the disease, the WGCNA R4.1.3 software package was used to conduct analyses. We screened for genes with missing values >5%, non-coding RNAs, and duplicated genes. Based on the scale-free topology criteria, the pickSoft Threshold function was used to select the appropriate soft threshold (beta) value and convert the neighbor-joining matrix to a topological overlap matrix. Modules were defined as the branches of a hierarchical clustering tree using topological overlap dissimilarity as input. Modules with similar expression profiles were merged into new modules with a fixed height of 0.25. In the correlation analysis of genetic significance and module membership in each module, genes within the modules showing a significant correlation with phenotypic traits (*p* < 0.05) were considered to be biologically relevant [30].

To identify differentially expressed genes (DEGs), we used the DESeq2 package (1.16.1) for RStudio (version 4.1.1) software. The *p*-values were adjusted using the Benjamini and Hochberg method. The corrected *p*-values and |log_2_foldchange| were used as thresholds for significant differential expression. Differentially expressed genes were considered candidates if the corrected *p*-value was less than 0.05. Genes from key modules were further analyzed for GO and KEGG pathway enrichment using the ClusterProfiler package (3.4.4). Protein–protein interaction (PPI) networks were constructed using the Cytoscape 3.10.2 software stringApp (v2.1.1) plugin. Hub genes were then screened using the cytoHubba (v0.1) plugin and were considered to be key module hub genes if |FoldChange| > 2, MMC > 5, and Degree > 5 [31].

### 2.9. Nissl Staining

Intact brains were removed at the end of the behavioral studies and immersed in 4% paraformaldehyde for 24 h. Following routine paraffin embedding, coronal sections were cut at a thickness of approximately 5 μm. These sections were deparaffinized through a graded ethanol series and stained with Nissl stain. They were sequentially washed in xylene, anhydrous ethanol, and 75% alcohol. Afterward, a differentiation solution was applied, using an acid alcohol solution (70% ethanol with 0.1% acetic acid) to remove excess stain. The slides were then mounted with neutral resin (Beyotime). The stained hippocampal sections were examined under a light microscope, and representative images were captured and analyzed, focusing on coordinates relative to bregma (AP: −1.3 to −2.9 mm). Microscopic examination was performed using a Nikon Eclipse CI microscope with 20x and 90x magnification.

### 2.10. Serum Corticosterone Measurement

Serum corticosterone (Cort) levels were quantified using a QuicKey Pro Mouse CORT ELISA Kit (Elabscience ELISA Kit, Wuhan, HB, China) following the manufacturer’s instructions. First, 50 μL of serially diluted standards (0–800 ng/mL), blank control (assay buffer), and serum samples were dispensed into the respective wells of the precoated antibody. After sample loading, 50 μL of HRP Conjugate working solution (Elabscience) was added to each well and incubated for 60 min at 37 °C. Following incubation, the decanted solution from each well, 350 μL of wash buffer (0.05% Tween-20 PBST), was added to each well. For signal development, 90 μL of Substrate Reagent (Elabscience) was added to each well and incubated for about 15 min at 37 °C. The enzymatic reaction was terminated by adding 50 μL of 2 M sulfuric acid stop solution, which induced a visible color transition from blue to yellow. The optical density (OD) was measured at 450 nm using a microplate reader (Thermo Fisher Scientific, Waltham, MA, USA), and Cort concentrations were determined by extrapolation from the standard curve generated with known concentrations of corticosterone standards.

### 2.11. RNA Extraction and Quantitative RT-PCR Detection

We assessed the reliability of the high-throughput sequencing results and the prediction of gene expression profiles via bioinformatics analyses. qRT-PCR was conducted to validate three differentially expressed genes, including the upregulated gene *Alb* and the downregulated genes *Npsr1* and *P2rx1*, all found in the CUMS+Noise group (Table 1).

**Table 1 biomedicines-13-01205-t001:** Hub gene primer sequence table.

Gene	Forward Primer Sequence	Reverse Primer Sequence
*Alb*	TGC TTT TTC CAG GGG TGT GTT	TTA CTT CCT GCA CTA ATT TGG CA
*Npsr1*	TGT GCC GAT GCT AGA TTC TTC C	CAG GAC CCA CAG GGT TAT CAG
*P2rx1*	GGA TGG TGC TGG TAC GAA ACA	CAC TGA CAC ACT GCT GAT AAG G

Total RNA was extracted using a HiPure Total RNA Mini Kit (Magen Biotechnology Co., Ltd., Guangzhou, China). After RNA extraction, the total RNA was first reverse-transcribed into cDNA, and then the cDNA was synthesized using a PrimeScript™ RT Reagent Kit with gDNA Eraser, followed by amplification with qPCR. Finally, the ABI 7300 Real-Time PCR System (Thermo Fisher Scientific, Waltham, MA, USA) was used for quantitative analysis. The comparative Ct method (2^–ΔΔCt) calculated the relative expression of each mRNA, and statistical significance was assessed using Student’s *t*-test [32].

### 2.12. Statistical Analysis

All of the results are expressed as the mean ± standard deviation. The results were statistically analyzed utilizing IBM SPSS Statistics version 19.0. Differences between groups were assessed using Student’s *t*-tests or one-way analysis of variance (ANOVA). Additionally, Kruskal–Wallis and Wilcoxon rank-sum tests were used to analyze the sleep recordings and scoring. Pearson’s correlation analysis was used to assess the association between key module hub genes and phenotypic information. Statistical significance was set at *p* < 0.05, and graphics were prepared by GraphPad Prism version 8.0.

## 3. Results

### 3.1. Mice Induced by CUMS and CUMS+Noise Showed Anxiety-like Behavior

In the open-field test (Figure 1B and Appendix A), the CUMS group exhibited a statistically significant reduction in TDMC% (*p* < 0.05), time spent in the center (*p* < 0.01) compared to the control group. In the CUMS+Noise group, only the time spent in the center (*p* < 0.01) was significantly decreased compared to the control group. Conversely, the CUMS group exhibited a statistically significant increase in TDMP% (*p* < 0.05), time spent in the periphery (*p* < 0.05) compared to the control group. In the CUMS+Noise group, only the time spent in the periphery (*p* < 0.05) was significantly increased compared to the control group. In the elevated plus maze test (Figure 1C and Appendix A), a significant reduction was observed in OAT% (*p* < 0.05) and OATDM% (*p* < 0.001) among the CUMS group in comparison to the control group. Additionally, the percentage of entries to the open arms (OAE%) exhibited a declining trend. In the CUMS+Noise group, both OAE% (*p* < 0.01), OAT% (*p* < 0.001), and OATDM% (*p* < 0.001) were significantly reduced compared to the control group. In both the tail suspension test and the forced swimming test, no significant differences in immobility duration were found between the CUMS and CUMS+Noise groups in comparison to the control group. The Cort concentrations are shown in Figure 1F and Appendix A. Compared to the control group, the Cort concentrations were significantly increased in both the CUMS group (*p* < 0.05) and the CUMS+Noise group (*p* < 0.01). Given the previously discussed results, both experimental mouse groups demonstrated characteristics of anxiety, while symptoms related to depression were not significantly present.

**Figure 1 biomedicines-13-01205-f001:**
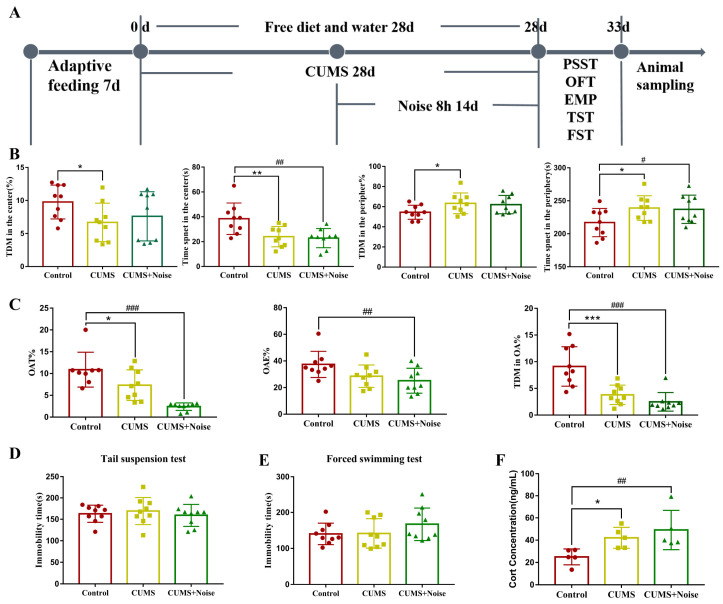
Anxiety-like behavior in the CUMS and CUMS+Noise model group mice. (**A**) Schematic diagram showing the experimental design and timeline (*n* = 9 per group). (**B**) In the open-field test (ANOVA, *n* = 9 per group), the observation indicators include the percentage of total distance moved in the center (TDMC%), time spent in the center (TPC), percentage of total distance moved in the periphery (TDMP%), and time spent in the periphery (TPP). (**C**) In the elevated plus maze test (ANOVA, *n* = 9 per group), the observation indicators include the percentage of time spent in the open arms (OAT%), the percentage of entries to the open arms (OAE%), and the percentage of total distance moved to the open arms (OATDM%). (**D**) In the tail suspension test, the immobility time was recorded (ANOVA, *n* = 9 per group). (**E**) In the forced swimming test, the immobility time was recorded (ANOVA, *n* = 9 per group). (**F**) Corticosterone concentrations (ANOVA, *n* = 5 per group). The results are expressed as the mean ± standard deviation and underwent statistical examination via ANOVA, following the successful validation through a normality test. * *p* < 0.05, ** *p* < 0.01, and *** *p* < 0.001, the CUMS group compared to the control group. ^#^ *p* < 0.05 ^##^, *p* < 0.01, and ^###^ *p* < 0.001, the CUMS+Noise group compared to the control group.

### 3.2. Sleep Disturbances Successfully Induced by the CUMS+Noise 8-H Method

We utilized the pentobarbital sodium-induced sleep test in conjunction with an advanced sleep recording and scoring system to thoroughly analyze the sleep patterns exhibited by the model mice (Figure 2 and Appendix A). The results of the pentobarbital sodium-induced sleep test are presented in Figure 2A. Notably, the sleep duration was significantly decreased in the CUMS+Noise group compared to the control group (*p* < 0.01). However, there were no significant differences in sleep latency found among the three groups. Subsequently, we used a computerized piezoelectric system to record sleep/wake activity during the dark phase (recording times: 8:00 p.m. to 8:00 a.m.) to observe more refined sleep patterns. As shown in Figure 2B and Appendix A, the percentage of sleep during the dark phase was significantly reduced exclusively in the CUMS+Noise group (*p* < 0.05) compared to the control group. During this period, sleep bout duration significantly decreased in the 20–21, 22–23, 27–28, and 29–30 time intervals in the CUMS+Noise group compared to the control group. Surprisingly, sleep bout duration significantly increased in the final 31–32 time interval (Figure 2C and Appendix A). In the analysis of single sleep duration percentages, the CUMS+ Noise group significantly decreased compared to the control group during the 120, 240, and 480 s intervals. It is noteworthy that the CUMS+Noise group exhibited a significant upregulation compared to the control group within the 1920 s interval (Figure 2C and Appendix A). These findings indicate that the mice in the CUMS+Noise group experienced disturbances in both sleep duration and circadian rhythm. However, these disturbances were not observed in the mice in the CUMS group.

Subsequently, the Nissl staining technique was conducted to investigate the pathological alterations in the hippocampus induced by different stressors. The findings are shown in Figure 2D. The structural condition of the hippocampal areas, particularly CA1, CA3, and DG, in the control group was found to be mostly normal. The significant characteristics observed were spherical nuclei and plentiful cytoplasm, complemented by a neatly structured and closely packed arrangement of neuronal cells visible in the area. The analysis of tissue histology indicated a tightly structured composition, lacking any indications of degeneration or necrosis. The hippocampal CA1, CA3, and DG regions in the model groups exhibited mild structural abnormalities. The observed abnormalities involved shifts in the configuration of neuronal cells across the visual field and signs of neuron degeneration, as indicated by the black arrows. This suggests that mouse hippocampal neurons in both model groups exhibited mild structural abnormalities.

**Figure 2 biomedicines-13-01205-f002:**
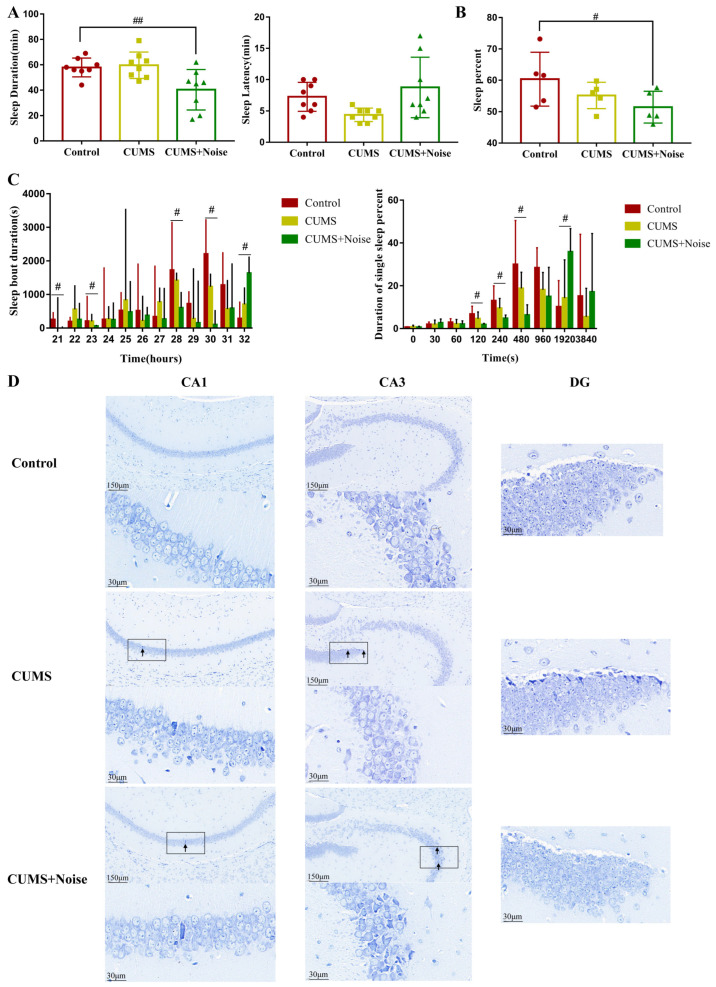
The CUMS+Noise mice showed sleep duration and circadian rhythm disturbances. (**A**) In the pentobarbital sodium-induced sleep test, sleep duration (ANOVA, *n* = 8 per group) and sleep latency (ANOVA, *n* = 8 per group) were recorded. (**B**) Sleep percentage in the dark phase (ANOVA, *n* = 5 per group). (**C**) Sleep bout duration (21 is the 8:00 p.m. to 9:00 p.m. time zone, 22 is the 9:00 p.m. to 10:00 p.m. time zone, etc.) and single sleep duration percentage in the dark, the results are expressed as the median ± IQR (Wilcoxon, *n* = 5 per group). (**D**) Hippocampus Nissl staining results at 20× and 90× magnification (Bar = 30 and 150 µm; *n* = 3 per group). # *p* < 0.05 and ## *p* < 0.01, the CUMS+Noise group compared to the control group.

### 3.3. Investigation of Hub Genes and Key Pathways Regulating CUMS in Conjunction with Noise Exposure Through WGCNA Analysis and Bioinformatics Approaches

#### 3.3.1. Key Gene Module Identification

We screened out genes with missing values >5%, non-coding RNAs, and duplicate genes. After preprocessing, 28,374 genes were found in the hippocampus. After screening, TPM data were log_2_-transformed and used. When R^2^ = 0.85 and Module Consolidation Threshold = 0.5, a scale-free network was formed, and we selected an optimal soft threshold of 13 in the hippocampus (Figure 3A). The gene modules identified by the dynamic clustering tree were marked with different colors, and the hippocampus was divided into 12 different module classes.

As shown in Figure 3B and Appendix A, the phenotypic information covered Cort concentrations, sleep duration (SD), time spent in the center (TPC), percentage of time spent in the open arms (OAT%), and the percentage of entries to the open arms (OAE%). Pearson’s correlation coefficients were utilized to assess the associations between the modules and the phenotypic information that displayed significant variations. This statistical approach facilitated a more comprehensive investigation into potential interactions among the phenotypes. A significant correlation was established if *p* < 0.05, along with R < −0.6 or R > 0.6. The analysis revealed that the brown module exhibited a significant and positive correlation with SD (*p* = 0.012, R = 0.75), TPC (*p* = 0.003, R = 0.83), OAT% (*p* = 0.042, R = 0.65), and OAE% (*p* = 0.003, R = 0.83). The brown module was negatively correlated with the Cort concentration; however, this difference was not significant. As a result, the brown module was recognized as a potential candidate module, playing a crucial role in the following analyses.

#### 3.3.2. Key Gene Module Expression and Gene Function

A clustered heatmap was used to analyze and display the gene expression profiles of the control and CUMS+Noise groups, focusing on the expression patterns of the brown module genes. Most of the genes were downregulated in the CUMS+Noise group compared to the control group (Figure 3C). This finding indicates that there were disparities in gene expression patterns between the two groups. Subsequently, a differential expression analysis was conducted on the genes within the brown module. Significant differences were considered when *p* < 0.05 and |FoldChange| > 2 (Figure 3D,E). Among them, 161 genes were significantly downregulated and 259 genes were significantly upregulated in the CUMS+Noise group compared to the control group. Following this, we performed an enrichment analysis on the genes in the brown module to explore the functional expression related to it.

A total of 2316 target genes were analyzed using RStudio (version 4.1.1) and the clusterProfiler package to conduct both KEGG pathway analysis and GO analysis. The application of a detailed analysis based on KEGG pathways led to the discovery of 96 pathways that were significantly enriched. The findings were subsequently visualized using RStudio (version 4.1.1) in conjunction with the dotplot functions in RStudio (Figure 3F and Appendix A). The GO analysis is similar in procedure to the KEGG analysis and consists of three parts: BP, MF, and CC. In Figure 3F and Appendix A, it is shown that GO analysis was significantly enriched in chemical synaptic transmission, positive regulation of synapse assembly, postsynaptic membrane, neuron projection, and GABAergic synapse. The target genes predicted by KEGG pathway analysis were significantly enriched in the glutamatergic synapse pathway, neuroactive ligand–receptor interaction pathway, axon guidance, circadian entrainment, GABAergic synapse pathway, and dopaminergic synapse pathway. Among them, we eliminated irrelevant disease pathways such as those related to morphine addiction and dilated cardiomyopathy. In summary, the gene expression function of the brown module is closely associated with mood and sleep disorders.

**Figure 3 biomedicines-13-01205-f003:**
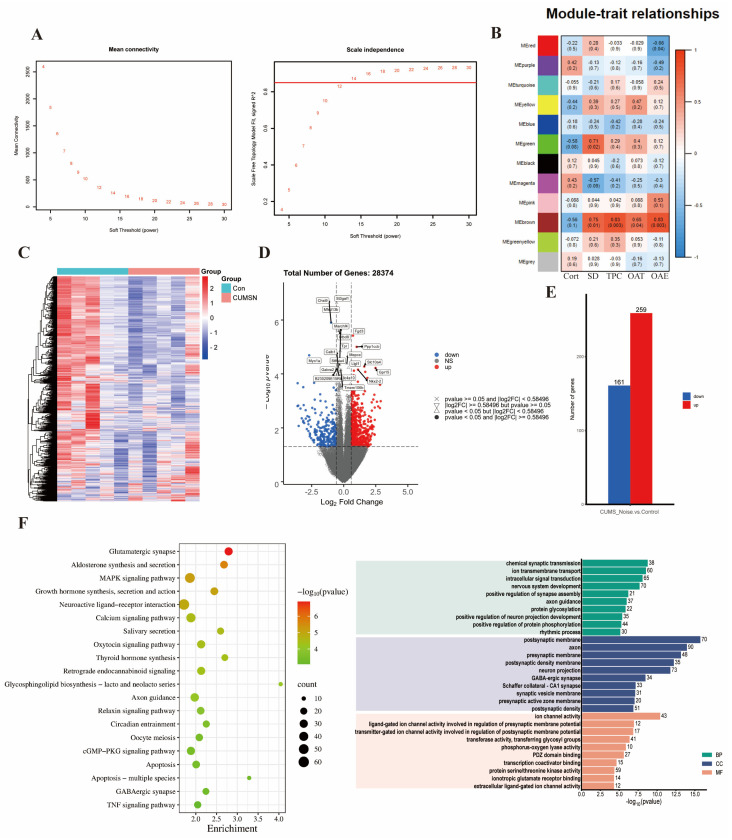
The key gene module expression function is closely related to mood and sleep disorders. (**A**) Soft thresholds calculated using the mean connectivity and scale-independent calculation methods. When R^2^ = 0.85 and Module Consolidation Threshold = 0.5, the soft threshold value is where the red horizontal line intersects the abscissa. (**B**) Heatmap of the module–trait relationship in the CUMS+Noise group compared to the control group. (**C**) Heatmap of gene clustering in the brown module of the CUMS+Noise group. (**D**) Volcanic plot of the brown module of the CUMS+Noise group. (**E**) Expression of shared differentially expressed genes in the control and CUMS+Noise groups. (**F**) KEGG and GO enrichment analysis.

#### 3.3.3. Selection and Verification of Key Module Hub Genes

Protein–protein interaction (PPI) networks were constructed using the Cytoscape 3.10.2 software stringApp (v2.1.1) plugin. Hub genes were then screened using the cytoHubba (v0.1) plugin, and a key gene was considered if MMC > 5 and Degree > 5 (Figure 4A). To further filter out key module hub genes, we took the intersection of the key genes, brown module genes, and differentially expressed genes (DEGs). The resulting 15 genes are shown in Figure 4B. These genes were considered key module hub genes and selected as the focus of this analysis; the results are shown in Table 2. We found that the hub gene *Alb* was significantly enriched in the thyroid hormone synthesis pathway, *Npsr1* was significantly enriched in the neuroactive ligand–receptor interaction pathway, and *P2rx1* was significantly enriched in the neuroactive ligand–receptor interaction and calcium signaling pathways. The hub genes *Rasl2-9* and *H2-Oa* showed significant enrichment within the pathway linked to human T-cell leukemia virus 1 infection. Subsequently, Pearson’s correlation analysis was conducted to assess the correlations between the key module hub genes and the phenotypic information that displayed significant differences. The results are shown in Figure 4C and Appendix A. *Alb* was significantly negatively correlated with TPC (*p* = 0.006, R = −0.79) and SD (*p* = 0.019, R = −0.72). *Npsr1* and *P2rx1* were significantly positively correlated with TPC (*p* = 0.017, R = 0.73 and *p* = 0.006, R = 0.80, respectively), SD (*p* = 0.049, R = 0.63 and *p* = 0.006, R = 0.80, respectively), and OAE% (*p* = 0.047, R = 0.64 and *p* = 0.049, R = 0.63, respectively). Notably, *Alb* was significantly negatively correlated with *P2rx1* (*p* = 0.015, R = −0.74), and *P2rx1* was positively correlated with *Npsr1* (*p* = 0.048, R = 0.64). These findings suggest that *Alb*, *P2rx1*, and *Npsr1* are all significantly associated with differential phenotypic information. Therefore, these hub genes will be chosen as the main targets for analysis in future PCR validation studies.

### 3.4. Validation of the Altered Expression of Hippocampal Genes in Mice Subjected to CUMS in Conjunction with Noise Exposure

We validated three functionally important hub genes using qRT-PCR. The expression levels of the validated hub genes are shown in Figure 4D and Appendix A. The differences in the expression of the *Alb* gene between the two groups were statistically significant. The results indicated a considerable upregulation of the hub gene *Alb* in the CUMS+Noise model mice, consistent with the findings from the high-throughput sequencing. This gene was significantly enriched in the thyroid hormone synthesis pathway in the KEGG analysis. It was also significantly enriched in the axon, neuron projection, GABAergic synapse, and long-term memory pathways, as well as in the regulation of long-term synaptic potentiation pathways, in the GO analysis. In conclusion, dysfunction of the key modular hub gene *Alb* may be an underlying molecular mechanism for sleep and mood disorders.

**Figure 4 biomedicines-13-01205-f004:**
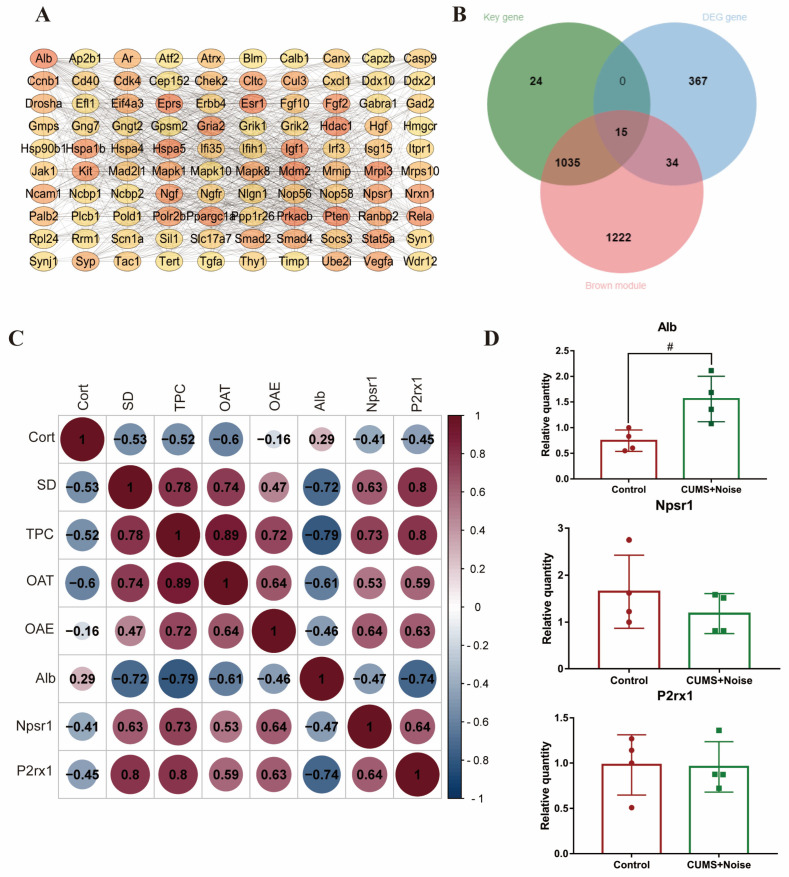
The key module hub genes significantly associated with differential phenotypes. (**A**) Candidate key gene PPI network diagram (only the top 100 portions of MMC > 5 and Degree > 5 are shown; detailed data can be found in Appendix A). (**B**) Venn diagram of DEG, key, and brown module genes. (**C**) Pearson’s correlation analysis of the key module hub genes with significantly different phenotypic information. (**D**) qRT-PCR validation results (Student’s *t*-test, *n* = 4 per group); the relative expression of each gene was calculated using the comparative Ct method (2^−ΔΔCt^). The results are presented as mean ± standard deviation, and statistical analysis was performed using Student’s *t*-test. # *p* < 0.05 compared to the control group.

**Table 2 biomedicines-13-01205-t002:** Key module hub gene information.

Gene Symbol	log_2_Fold Change	*p*-Value	Degree	MCC	Pathway
*Alb*	1.974	0.004	115	761,150,689	Thyroid hormone synthesis
*Npsr1*	−1.495	0.009	65	111,224,013,584	Neuroactive ligand–receptor interaction
*Rasl2-9*	1.217	0.047	14	26,047	Human T-cell leukemia virus 1 infection
*Ccl22*	1.420	0.016	12	11,569	None
*H2-Oa*	1.041	0.040	11	728	Human T-cell leukemia virus 1 infection
*Pou4f1*	1.715	0.023	11	133	None
*Reg3g*	−2.645	0.021	10	24	None
*Qprt*	1.047	0.029	9	12	None
*Actg2*	−1.655	0.020	8	37	None
*Slc45a2*	1.266	0.004	8	14	None
*Wnt8b*	1.387	0.035	8	8	None
*Abca12*	−1.493	0.011	7	19	None
*Cnga3*	1.317	0.005	7	15	None
*P2rx1*	−1.168	0.005	7	14	Neuroactive ligand–receptor interaction; Calcium signaling pathway
*Serpina1b*	1.434	0.023	7	7	None

Note: Key module genes were considered hub genes when they conformed to |FoldChange| > 2, *p* < 0.05, MMC > 5, and Degree > 5.

## 4. Discussion

Stress can be characterized as a natural and adaptive reaction to external pressures. However, prolonged exposure to high levels of stress can lead to sleep-onset insomnia, short-term insomnia, and mood disorders [33]. An increasing amount of research suggests that stress-induced insomnia and mood disorders (e.g., anxiety and depressive disorders) influence each other. Research has shown that stress-induced insomnia and anxiety–depressive disorders share a common neural circuit—a complex network of receptor systems including dopamine, serotonin, and adenosine. These also serve as the physiological foundation for both arousal and sleep [34,35]. Currently, the main methods used to establish sleep deprivation in rodent models include horizontal stage deprivation, stress deprivation, chemical agent deprivation, mild stimulus deprivation, and forced movement deprivation [36]. Among these models, we suggest that the stress-induced animal model is more consistent with the prevalent type of insomnia observed in humans. However, the specific details of model preparation and regulatory mechanisms are not fully understood. Therefore, we used CUMS in combination with sleep deprivation to establish a stress-induced insomnia model. The evaluation of sleep state and mood alterations in the mice was carried out using the sleep recording and scoring system, the pentobarbital sodium-induced sleep test, and behavioral tests. The results indicated that the model mice had disturbed circadian rhythms and durations, along with manifestations of anxiety-like behavior. To further explore the mechanism of interaction between stress-induced mood disorders and sleep disorders, we conducted an analysis of hippocampal high-throughput sequencing data to identify key genes using the WGCNA method. Finally, PCR experiments were performed to verify the significantly different hub genes. Given previous results reported in the literature, this study was conducted to better understand the possible connections between sleep and mood disorders.

Our research demonstrated that the CUMS+Noise 8 h intervention method successfully established a model for stress-related insomnia. The sleep disturbances were characterized by a reduction in sleep duration and irregularities in circadian rhythm. Circadian rhythm disturbances in the mouse model group were specifically characterized by a significant decrease in multiple sleep time intervals and single sleep duration, and a significant increase in sleep duration at the final time interval [27]. Previous studies have assessed sleep duration based on REM and NREM test results [13,14]. The results of this research contribute additional insights and improvements to the current literature. In particular, sleep interval duration was significantly upregulated in the CUMS+Noise group at the final time interval of ZT23-24 and at the 1920 s time interval (percentage of single sleep duration) compared to the control group. The regulatory effects of sleep homeostasis may be responsible for this phenomenon, which manifests itself as sleep rebound after prolonged wakefulness [37]. The CUMS+Noise eight-hour method was equally successful in inducing anxiety-like behavior. Furthermore, our findings indicate that the four-week CUMS method did not induce sleep disturbances. Nissl staining results revealed mild structural abnormalities in the CA1 and CA3 or dentate gyrus (DG) regions of the hippocampus in both model groups. These results suggest that while the CUMS protocol did not affect sleep patterns, it did lead to observable changes in hippocampal structure. Specifically, neurons were systematically arranged in the visual field, and a small amount of neuronal degeneration was observed. This finding indicates that the CUMS+Noise eight-hour method can successfully establish a model of stress-induced insomnia.

Subsequently, we conducted an analysis of the hub genes present in the hippocampal high-throughput sequencing data using the WGCNA method. The brown module hub genes *Alb*, *P2rx1*, and *Npsr1* were found to be significantly associated with differential phenotypes, including Cort concentration, SD, TPC, and OAE%. The results of the PCR experiments showed that only *Alb* was significantly different between the two groups. Albumin is known to be one of the transport proteins for thyroxine (T4) and has a strong binding affinity for T4 [38]. These findings imply that the interaction between sleep disorders and anxiety-like behavior may be closely linked to the dysfunctional transportation of thyroid hormones by albumin. The significant elevation of brain albumin observed in this study may suggest increased levels of thyroid hormones in the CUMS+Noise group. Proper thyroid hormone levels are vital for healthy neurological activity, and extended periods of chronic stress can affect the thyroid hormone signaling pathway [39]. Excessive thyroid hormone exposure exacerbates neuronal death and reduces brain volume, affecting memory, concentration, mood, vision, and motor function [40]. There is a strong correlation between thyroid hormones and sleep duration, with longer sleep being linked to significantly lower levels of peripheral thyroid hormones [41]. Research has shown that alterations in thyroid hormone metabolism occur due to changes in the functioning of the lower hypothalamic–pituitary–thyroid (HPT) axis as a result of sleep deprivation. This adaptation is characterized by an elevation of peripheral T3 levels, which occurs through enhanced conversion of T4 to T3 in peripheral tissues [42]. High levels of peripheral T3/T4 levels affect Aβ turnover in the brain by altering the blood–brain barrier and the ratio of receptors in glial cells. This modulation subsequently impacts the dynamics of Aβ turnover within the central nervous system [43]. Thyroid hormones also play an important role in mood regulation. Research has demonstrated that thyroid hormone receptor alpha 1-knockout animals show higher levels of anxiety, while beta-knockout males display lower levels of anxiety [44]. Another study found that insomnia symptoms with major depression may be associated with higher body mass index and waist circumference and lower levels of metabolic syndrome components, triglycerides, insulin, and albumin [45]. The previous study indicated that thyroid hormones could be essential in managing sleep and mood disorders that are linked to stress. In particular, serum albumin possesses a tryptophan residue at the amino acid position 214, and the interaction between albumin and thyroxine significantly influences the interaction between albumin and tryptophan. Albumin affects the production of serotonin in the brain by binding free tryptophan in plasma, which in turn regulates its transportation into the central nervous system [46]. Serotonin plays an active role in the regulation of sleep architecture through 5-HT2A receptors [47]. Further research is needed to explore how albumin affects the production of serotonin in relation to thyroid hormone interactions during stress-induced insomnia disorder. Therefore, the underlying mechanism of the interaction between sleep disorders and anxiety-like behavior may be closely related to dysfunctional albumin transport of thyroid hormones. This study aimed to investigate the feasibility of a model protocol for stress-induced insomnia and examine the potential biological mechanisms underlying the correlation between mood and sleep. Consequently, the sample size was limited to ten animals per group, and the preliminary observations necessitate further experimental validation.

## 5. Conclusions

The CUMS+Noise eight-hour intervention successfully induced a model of stress-induced insomnia with anxiety-like behavior. The sleep disturbances observed in the mice were specifically characterized by sleep duration and circadian rhythm disturbances. Stress-induced sleep disorders and mood disorders are closely associated with upregulation of *Alb*; however, the specific pathophysiological mechanisms need to be further investigated.

## Data Availability

The original contributions presented in the study are included in the article; further inquiries can be directed at the corresponding authors.

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
