# Peer review of "Hippocampal Transcriptome Analysis in a Mouse Model of Chronic Unpredictable Stress Insomnia"

_biomedicines, 2025, doi:10.3390/biomedicines13051205_

Round 1
Reviewer 1 Report
Comments and Suggestions for Authors
Abstract:
- "Anxiety-like mood disturbances occurred in model groups". You mention anxiety-like mood disturbances, but anxiety was only evaluated through behavioral measures (open field test, elevated plus maze).
- "Sleep disturbances... characterized by disruptions in sleep duration and sleep rhythm”. The term sleep rhythm is not very specific. If you are referring to the circadian rhythm, it should be stated explicitly."
- "Significant increase in sleep duration at the final time interval". It is unclear what is meant by the 'final time interval.' Is it a specific time period? A recording phase? Could you clarify?
- "Hub genes... Alb, P2rx1, Npsr1". That's fine, but when you mention genes associated with key phenotypic characteristics, you don't clarify how that association was determined. Was it correlation? WGCNA + modules + trait correlation?
5."Albumin... is significantly enriched in the thyroid hormone secretion pathway". Stating that 'albumin is enriched in the pathway' might be confusing. In reality, it is a transport protein, not a direct effector of the thyroid axis.
- "Thyroid hormone dysfunction" like mechanism. Although thyroid dysfunction may be involved, only differences in the expression of Alb were found, which is a transporter, not a thyroid hormone or its receptor. Please modify.
Introduction
In general, the introduction is well described. However, several revisions are recommended to enhance clarity and precision:
1.The terms “long-term chronic stress” and “prolonged exposure to chronic stress” are redundant, as "chronic" already implies a long duration. Consider using either “chronic stress” or “prolonged exposure to stress.”
2.The structure of the phrase “may lead to the development of stress-related insomnia into chronic insomnia” is confusing. Suggested alternatives include:
“…may lead to the progression of stress-related insomnia into chronic insomnia”
“…may cause stress-related insomnia to become chronic insomnia”
- The phrase “Sleep disorders of this nature are typically characterized by…” is somewhat ambiguous. Replace with: “These sleep disturbances are typically characterized by…”
4.Correct the typo “teh” to “the.”
5.Replace “and so on” with a more formal alternative: “among others.”
6.The phrase “the preparation method utilizing the combined CUMS/CRS model…” is confusing. A clearer alternative would be: “The combined CUMS/CRS model with sleep deprivation is considered to more accurately replicate clinical symptoms of stress-induced insomnia.”
7.Replace “behaviour texts” with: “behavioral tests” or “behavioral assessments.”
- The phrase “sleep-wake scoring and recordings data” could be more accurately stated as:
“sleep-wake cycle scoring and electrophysiological recordings” or “sleep architecture data.”
9.Lastly, consider replacing “mice from the model group” with the more concise: “model mice.”
Materials and Methods
Animals:
1.The phrase “6–8 week SPF male C57BL/6J mice” should be revised for clarity and completeness. Suggested replacement: “Specific pathogen-free (SPF) male C57BL/6J mice, aged 6–8 weeks and weighing 20 ± 2 g...”
- Add when the animals’ body weight were measured.
- In lists of three or more terms, commas should be used consistently for clarity. Replace:
“Control, CUMS model and CUMS+Noise model groups” with: “Control, CUMS model, and CUMS+Noise model groups.”
4.. Replace time notations “ZT0:00–12:00” and “ZT12:00–24:00” with the more concise and standard format: “ZT0–12” and “ZT12–24.”
Instruments and Reagents:
1.Avoid repetition of provider names across the equipment list unless necessary for clarity. Group items from the same manufacturer together when possible.
- Group similar equipment by category, such as behavioral equipment, molecular analysis tools, etc., to enhance readability and organization.
3.Replace “Nuldus” with the correct spelling: “Noldus.”
4.Replace “Forced Swimming Bucket” with more precise terminology, such as:
“forced swim test apparatus” or “forced swim test device.”
CUMS and CUMS+Noise model:
1.Improve the first sentences, as they are currently split in an unnatural way. Consider combining them into a single, well-structured sentence with appropriate punctuation to enhance readability.
2.Replace “behavioral texts” with the correct term: “behavioral tests.”
3.Replace “Eyeball blood collection method” with a more accurate and professional expression: “Blood was collected via retro-orbital sampling under anesthesia.”
Pentobarbital sodium sleep test (PSST):
1.Avoid unnecessary repetitions of the word “sleep.” Revise sentences to maintain clarity without overusing the term.
- Include a brief explanation to clarify terminology: “The righting reflex is a commonly used parameter to assess sleep onset in rodent models.”
Behavioral tests:
1.All behavioral tests must include a habituation period to the experimental room to minimize stress and variability in the animals’ responses. While you added this information in the open field test description, it is absent from the other behavioral test descriptions. Please ensure that each behavioral test description includes the habituation procedure.
2.To ensure methodological consistency and reproducibility, all behavioral tests were preceded by a habituation period in the experimental room. Equipment was cleaned between each animal using 75% ethanol to eliminate olfactory cues. Behavioral sessions were recorded and analyzed using [insert name of software, e.g., EthoVision XT, Noldus Information Technology]
- Please indicate the time of day at which the behavioral tests were conducted (e.g., during the light or dark phase, or specific ZT times), as this can significantly affect the outcomes due to the circadian activity patterns of rodents.
“Open field text(OFT)” must be replaced by “Open field test (OFT)”
1.The phrase “The open field box was performed..." is incorrect, the tests are performed, no the boxes. Replace by “The open field test was conducted...".
- Number of animals at the same time: saying that 3 mice (2 model and 1 control) were placed simultaneously in different boxes is not entirely clear. It's better to explain it more precisely. Each animal was placed individually, right?
- "Nuldus" must be replaced by “Noldus”
Considering that the sand in the device can be virtually divided into 3 zones (center, intermediate, and periphery), it would be appropriate to include a figure indicating the dimensions of the central zone.
Elevated plus maze test(EPM):
- Redundancy: The description of the EPM structure is repeated twice.
2.It does not mention the height of the closed walls.
3."At the beginning of the text" must be replaced by "At the beginning of the test."
- Replace "Central open area" by "center" or "central area."
5.The description of the "preference for open arms" must be improved. "Preference for being in open arms..." should be replaced by "percentage of entries into open arms and time spent there."
Tail suspension test(TST)
1."had their tails suspended by a restraint band located 2 cm from the tail's end" should be replaced by "were suspended by the tail using adhesive tape placed 2 cm from the tip."
2."individuals were placed..." should be replaced by "mice were placed..."
3.Eliminate the unnecessary repetition of "immobility."
4."delineates the duration..." sounds confusing and should be improved. Consider replacing it with "defines the duration..." or "indicates the duration..." depending on the context.
Forced swimming test(FST):
1.Redundancy in the definition of immobility: it is explained twice with similar phrases. You can remove the repetition and consolidate the explanation in one clear sentence.
2."Passive suspension" doesn't fit here (it applies well in TST). It should be replaced by "minimal movements necessary to keep the head above water."
3.Replace "Positioned within a Plexiglas cylinder" with "placed in a Plexiglas cylinder."
4.Add water height: if available, ideally around 15 cm, to prevent the mice from touching the bottom.
5.Replace "analysed" with "analyzed."
6.Logical order: first describe the test, then the analysis.
Sleep recordings and scoring:
1.Confusing or contradictory phrase: "Automatic scoring > 0 is sleep and automatic scoring < 0 is sleep." This sentence is incorrect as both conditions cannot represent "sleep." One should likely refer to "wake" and the other to "sleep." The phrase should be revised to: "Automatic scoring > 0 represents wake, and automatic scoring < 0 represents sleep," assuming that’s the intended logic. If this is not the case, you might need to check for a typographical error.
2.Incorrect or exaggerated statement: "Piezoelectric systems have been validated to be up to 90% more accurate than EEG/EMG-based sleep recording." This is incorrect. Piezoelectric systems have been validated to show up to ~90% concordance with EEG/EMG, but they are not more accurate. The statement should be modified to: "Piezoelectric systems have been validated to show up to ~90% concordance with EEG/EMG-based sleep recordings, but EEG/EMG remains the gold standard."
3.Non-standard or confusing terms: The terms "array duration" and "arousal array duration" are not commonly used in this context. It is better to clarify that you are referring to the duration of wake or sleep episodes. For example: "The duration of wake episodes" or "The duration of sleep episodes." The term "average array duration" is ambiguous and should be rephrased as "the average duration of wake or sleep episodes."
- "The detection time is the 12h dark phase, ZT12:00–ZT24:00, i.e. 20:00 to 8:00." There is a time discrepancy. ZT12:00 to ZT24:00 is from 20:00 to 8:00 only in an inverted cycle, which hasn’t been clarified. The time description should be revised for clarity: "The detection time occurs during the 12-hour dark phase, ZT12:00–ZT24:00, corresponding to 20:00 to 8:00 in a standard cycle where ZT0 = lights on at 8:00 and ZT12 = lights off at 20:00." Furthermore, if sleep detection was performed during the dark phase, this is a mistake because mice are nocturnal animals.
RNA sequencing:
1.Confusing phrase: "specific chemicals to ensure that only RNA was used as template." This is vague. A clearer way to describe the process would be: "Random hexamer primers and reverse transcriptase were used to synthesize cDNA from RNA, and inhibitors of DNA contamination were included to prevent the use of DNA as a template."
2.Clarification on estimated size: "The estimated size is 250–300 bp." While this is likely referring to the size of the final fragment of the library, it should be clarified further: "The estimated size of the final PCR product is between 250–300 bp."
3.Redundant phrase: "The PCR product is purified again..." The word "again" is redundant if it hasn't been mentioned before. It can be rewritten as: "The PCR product is purified," or, if this is the second purification, "The PCR product is purified a second time..."
4.Informal wording: "the libraries have been checked and approved." This sounds informal and lacks detail. A more formal and accurate description would be: "The quality and quantity of the libraries were assessed using Bioanalyzer or qPCR, and subsequently, the libraries were pooled."
WGCNA analysis and key module bioinformatics analysis:
1.First-person plural: To avoid the loss of fluency caused by the first-person plural, revise the text to maintain a more neutral tone. For example, change: "We conducted the analysis..." to "The analysis was conducted..."
2."Interactions between genes and disease-differentiating phenotypes": Replace with: "Gene modules associated with phenotypic traits related to disease," which is clearer and more technical, especially in the context of WGCNA (Weighted Gene Co-expression Network Analysis).
3."Disease-differentiating phenotypes": This term is ambiguous. You can improve it by specifying the context more clearly. A better version could be: "Disease-associated phenotypic traits" or "Phenotypes that differentiate disease states."
4."Modules are then clustered and similar modules are combined": This can be made more precise by changing it to: "Modules with similar expression profiles were merged."
5."P < 0.05 indicated that the genes in the module were consistent with the sample traits": A clearer and more scientifically appropriate version would be: "Genes within modules showing a significant correlation with phenotypic traits (P < 0.05) were considered to be biologically relevant."
6."Heterogeneous expression of candidate genes in both groups was analysed": This phrase is vague. Replace it with: "To identify differentially expressed genes (DEGs), we used..." This ensures clarity and precision.
- "DEGs": Include the term "DEGs" after introducing "differentially expressed genes" to maintain consistency. For example: "To identify differentially expressed genes (DEGs), we used..."
8."Key module genes were then analysed for GO and KEGG…": To improve fluency and avoid redundancy, replace with: "Genes from key modules were further analyzed for GO and KEGG..."
9."stringApp (v2.1.1) plug-in": Use the more standard phrasing: "stringApp plugin (v2.1.1)."
Nissl staining:
1.Atlas and Coordinates: Include which anatomical atlas was used for the brain sections. For example: "Brain sections were performed according to the [specific atlas name] and coordinates relative to bregma were used (e.g., AP: +1.5 mm, or as appropriate for the region of interest)."
2.Replace “behavioural” with “behavioral” to follow standard American English spelling.
3.To follow a more standard scientific notation, replace “approximately” with the shorthand:
“…~5 μm..”
4.“Gradient deparaffinisation was followed by Nissl staining” replace with: “Sections were deparaffinized through a graded ethanol series and stained with Nissl stain.”
5.“Remaining sections”: Clarify the meaning of “remaining sections” to avoid confusion. You can revise: “The sections were then treated…” if you are referring to all sections. If referring to specific ones, consider specifying what happened to others (e.g., “The sections that were not used for staining were stored for further analysis”).
6.“Differentiation solution”: Be more specific about the composition of the differentiation solution. For example: “Differentiation was carried out using an acid alcohol solution (e.g., 70% ethanol with 0.1% acetic acid) to remove excess stain.”
7.“Sealed with neutral adhesive” Replace with: “Mounted with neutral resin” (e.g., Permount or similar). This is more specific and standard in histology.
8.“Neutral adhesive”: As mentioned, replace with “neutral resin” or specify the brand if known (e.g., Permount, DPX).
- “Finally, microscopic examination was performed and images were captured and analysed”: This can be made more precise by revising to: “Finally, stained sections were examined under a light microscope, and representative images were captured and analyzed.”
- Microscopic Examination Details: If possible, add further detail about the microscope used, such as: “Microscopic examination was performed using a [type of microscope] with [magnification, e.g., 10x, 40x] magnification.”
RNA extraction and quantitative RT-PCR detection:
- "authenticity of the results" replace with: "reliability," "accuracy," or "validity" depending on the specific context. For example, "We aimed to assess the reliability of the results."
- “high-throughput detection": Replace with: "high-throughput sequencing" to properly describe RNA-seq or similar methods.
3.“the prediction of mRNA expression by bioinformatics analysis": Replace with: "the prediction of gene expression profiles by bioinformatics analyses" for more precision and correct plural usage.
4.“verify differentially 3 expressed mRNAs": Replace with: "validate three differentially expressed mRNAs" to correct the grammatical error and use the appropriate verb in this context.
5.“including the up-regulated mRNA in the CUMS+Noise group: Alb, and the down-regulated mRNAs in the CUMS+Noise group: Npsr1 and P2rx1." Replace with: "including the upregulated gene Alb, and the downregulated genes Npsr1 and P2rx1, all in the CUMS+Noise group." This is more precise as it refers to genes, not mRNA, and avoids repetition of "CUMS+Noise group."
6.“cDNA was amplified using...": Modify as: "cDNA was synthesized using [kit name], followed by amplification with qPCR." This distinction clarifies that amplification occurs during the qPCR step, not during cDNA synthesis.
7.“ABI 7300 real-time qPCR system": Replace with: "ABI 7300 Real-Time PCR System" for correct capitalization.
8.“comparative CT method (2-ΔΔCT)": Replace with: "comparative Ct method (2^–ΔΔCt)" to correct the notation for the method used in qPCR analysis.
9.“and t-test was used to evaluate the results":Replace with: "and statistical significance was assessed using a Student’s t-test" for clarity and proper scientific phrasing.
Statistical analysis
- “mean±standard deviation": Correct format: "mean ± standard deviation" with spaces before and after the "±" symbol, as per typographic conventions. Also, SD abbreviation is also used with sleep deprivation.
2.“Differences between groups were determined by...": Replace with: "Differences between groups were assessed using..." This is more common and technically correct in this context.
3."Kruskal Wallis": Corrected to: "Kruskal–Wallis" (with an en dash between "Kruskal" and "Wallis").
4."Wilcoxon tests": Specify the type of test as: "Wilcoxon rank-sum tests" to indicate that this is the non-parametric equivalent of the t-test.
5.“relationship”: Replace with: "association" or "correlation," as "relationship" is too informal in scientific writing.
6.“key modular hub genes": Replace with: "key module hub genes" for accuracy, as "modular" is incorrect in the context of WGCNA. The term "module hub genes" is the correct terminology.
7."signifcance": Correct the typo: "significance."
Results
1.In general, the values of mean ± SD are included both in the main text and in the supplementary tables. This redundancy makes the manuscript difficult to read. You should indicate only the statistical differences in the main text and refer to the corresponding figures and supplementary tables for detailed values.
- The analysis of body weight variation over time presents methodological inconsistencies. First, the results include periods during which food deprivation was applied as a stressor, which could confound the interpretation of body weight changes. Proper assessment of this parameter should include body weight measurements taken at the endpoint, after the stress protocol has been completed. Additionally, body weight measurements collected over time should be analyzed using repeated-measures ANOVA, which allows for accurate evaluation of time-dependent effects and potential group-by-time interactions. Only after confirming a significant interaction or a main effect of the experimental condition can it be concluded that the model influences body weight. Furthermore, food intake should be monitored throughout the experimental period to determine whether the model affects feeding behavior or whether other physiological mechanisms are involved—especially considering that the food deprivation stressor could have a direct impact on this parameter.
3.In the Open Field Test (OFT), the CUMS group showed a statistically significant reduction in both total distance moved and number of crossed lines. These findings suggest decreased locomotor activity but do not necessarily indicate increased anxiety-like behavior. To appropriately assess anxiety, the analysis should be normalized. While you mention mean velocity and time spent in the center, a more rigorous approach would include quantifying the distance traveled, number of entries, and time spent in three virtual zones (central, intermediate, and peripheral). These parameters should then be normalized to the total distance traveled or total entries, as appropriate.
- In the Elevated Plus Maze (EPM), total distance traveled must also be reported. If differences in this parameter are observed, all other behavioral measures (e.g., entries and time spent in open/closed arms) should be normalized accordingly. Consider including the distance traveled specifically in the open arms. Additionally, the abbreviations used for arm-related parameters are unclear. It is recommended to adopt standard terminology such as OAE (Open Arm Entries), OAT (Open Arm Time), and OAD (Open Arm Distance).
- The time spent in the open arms is presented as one of the evaluated parameters; however, this was not mentioned in the Methods section. There, it is stated that “the preference for being in open arms over closed arms (expressed as a percentage of entries to open arms) was analyzed.” This description must be expanded to include both the number of entries and the time spent in the open arms, as both are standard and complementary measures of anxiety-like behavior in the elevated plus maze.
5.In addition, the methodology for corticosterone determination is entirely missing from the Methods section. It is essential to indicate how corticosterone was measured (e.g., ELISA, RIA), whether animals were fasted prior to sampling, and at what time of day euthanasia was performed, as these factors can significantly influence hormone levels.
6.Regarding Figure 1, the legend should be revised to include the standard abbreviations (e.g., OAT for open arm time, OAE for open arm entries), which facilitates clarity and consistency with the rest of the manuscript.
7.On Sleep disturbances successfully induced by CUMS+ Noise 8h method:
-It is unclear whether the measurements presented in Figure 2C are repeated measures. If so, the data should be analyzed using a repeated measures ANOVA, as this statistical test is appropriate for accounting for within-subject variability over time or conditions.
-Use “..exhibited by the mice” instead of “displayed by the mice”
-Replace “pentobarbital sodium sleep test” by “pentobarbital sodium-induced sleep test”.
-Replace “markedly increased” by “significantly increased”
-Replace “notable statistical differences” by “significant differences”
-Replace “computerised” by “computerized”
-Replace “percentage of dark phase sleep” by “percentage of sleep during the dark phase”
-Replace “was found to be significantly reduced” by “was significantly reduced”
-The terms “down-regulated” and “up-regulated” are generally used in the context of gene expression and are not appropriate for describing changes in behavioral parameters such as sleep episode duration. These should be replaced by “decreased” and “increased,” respectively, to accurately reflect the nature of the observed changes in behavior.
-Replace “stresses” by “stressors”
-Replace “showed” by “shown”
-Figure 2 legend: which is the microscope magnification used in D images?
- On Key gene module identification:
- Replace “After pretreatment” by “After preprocessing”. In bioinformatics, “preprocessing” is a more appropriate term than “pretreatment.”
-Replace “variables” by “phenotypes” / “traits”
-Replace “TPM data... was analyzed and log2 was calculated” by “TPM data were logâ‚‚-transformed and used...” This adjustment improves the accuracy, as it refers to a transformation, not a simple calculation, and ensures proper concordance ("data" is plural in scientific English).
- On Key gene module expression and gene function:
-Remove “obviously”. Removing "obviously" makes the tone more formal and appropriate for scientific writing.
-Replace “DOTPLOT software packages” by “dotplot functions in RStudio”. This adjustment clarifies that it's a function from the clusterProfiler package, not a separate software.
- Consistency between analysis names and GO categories (BP, MF, CC).
10.On Selection and verification of key module hub genes
-Consistency in gene and pathway names: Genes (e.g., Alb, P2rx1, etc.) should be italicized, and official names for pathways should be used.
-Grammar accuracy: For example, “The findings is shown” should be corrected to “The resulting 15 genes… are shown”.
-Logical clarity: rearrange the paragraphs to improve the natural progression: selection → functions → correlations → conclusions.
-Academic style: remove repetitions and improved connectors.
- On Validation of the altered expression of hippocampal genes in mice subjected to CUMS in conjunction with noise exposure
- Consistency in style: Gene names should be italicized, pathways in lowercase without quotation marks, and "GO" and "KEGG" should be used as proper nouns.
-Elimination of repetitions: For example, “The gene was significantly enriched…” repeated twice consecutively.
-More precise conclusion: add “molecular mechanisms underlying…” to reinforce the functional impact of the finding.
Discussion
- (Line 481–482). sleep-onset insomnia, short-term sleep insomnia. " Short-term sleep insomnia" sounds redundant or unclear. It is recommended to use: "transient insomnia" or simply "short-term insomnia."
- (Line 483–486). These also serve as the physiological foundation..." You correctly identified the subject-verb agreement issue — "These" is plural, so the verb must be "serve" instead of "serves".
- (Line 491–492). The specific details of model preparation and regulatory mechanisms are not fully understood.
- (Line 497). “..manifestations of anxiety-like mood disturbances”. Slightly redundant (mood disturbances and anxiety-like).
- (Line 511–514). “interval 31–32h and at time interval 1920s” Confusing: Is "1920s" a typo? Maybe it should be "interval 19–20 h" or "1920 s (seconds)"
Conclusión
Errores detectados y sugerencias de corrección:
Lexical redundancy: "sleep disturbances were specifically characterized by sleep duration and sleep rhythm disturbances". “Sleep disturbances” is repeated twice in the same sentence, creating a feeling of unnecessary repetition.
Supplementary material
1.Add units in the tittle tables
2.Add F values in the table description followed of p values. Remove them from the tables.
3.There are several mistakes like “text” instead of “test” and typing error.
4.Do not repite the ANOVA description every time you mention it.
Comments on the Quality of English Language
The manuscript addresses a relevant topic; however, I strongly recommend a thorough revision of the English writing. There is an inconsistent use of passive and active voice, variations between first-person singular and plural expressions, and several grammatical errors throughout the text. Additionally, the manuscript shows a mixture of British and American English, which further affects the overall clarity and consistency.
Author Response
Responses to Reviewers' comments
Dear Reviewers,
We are very grateful for your comments on our manuscript entitled "Hippocampal transcriptome analysis in a mouse model of chronic unpredictable stress insomnia" (Biomedicines-3521469). These comments have been very valuable in helping us revise and improve the manuscript. It has also made us realize that there are problems with English writing and the use of statistical methods. We have carefully considered these comments and revised the manuscript accordingly. According to the submission guidelines and your suggestions, we have carefully revised the manuscript and highlighted the revised parts in red in the text. The major revisions and responses to the reviewers' comments are listed below.
Responses
Abstract
- "Anxiety-like mood disturbances occurred in model groups". You mention anxiety-like mood disturbances, but anxiety was only evaluated through behavioral measures (open field test, elevated plus maze).
- Weapologize for incorrectly equating anxiety-like behavior with mood disorders. We have replaced "anxiety-like mood disturbances" with "anxiety-like behavior" in the texton lines 18, 32, 309, 334, 535, 555, 602 and 610.
- "Sleep disturbances... characterized by disruptions in sleep duration and sleep rhythm". The term sleep rhythm is not very specific. If you are referring to the circadian rhythm, it should be stated explicitly."
- We agree that the reference to "sleep rhythm" is unclear. Therefore,we have replaced"sleep rhythm" with "circadian rhythm" throughout the manuscript on lines 21, 368, 384, 534, 544, 545, and 611.
- "Significant increase in sleep duration at the final time interval". It is unclear what is meant by the 'final time interval.' Is it a specific time period? A recording phase? Could you clarify?
- Weapologize for using "at the final time interval" in an unclear sense. We have replaced "at the final time interval" with "at the final time interval of ZT23-24" in the text on lines 23, 24 and 551.
- "Hub genes... Alb, P2rx1, Npsr1". That's fine, but when you mention genes associated with key phenotypic characteristics, you don't clarify how that association was determined. Was it correlation? WGCNA + modules + trait correlation?
- We used the Pearson correlation analysis method to determine the correlation between key phenotypes and hub genes. We have revised the manuscript online26.
- "Albumin... is significantly enriched in the thyroid hormone secretion pathway". Stating that 'albumin is enriched in the pathway' might be confusing. In reality, it is a transport protein, not a direct effector of the thyroid axis.
"Thyroid hormone dysfunction" like mechanism. Although thyroid dysfunction may be involved, only differences in the expression of Alb were found, which is a transporter, not a thyroid hormone or its receptor. Please modify.
- We agree with your comments about the function of Albumin.We have made the following revisions "In conclusion, the underlying mechanism for the interaction between sleep disorders and anxiety-like behavior may be closely related to the dysfunctional albumin transportation of thyroid hormones."It's in the text on lines 31-33, and 601-603.
Instruction
- The terms "long-term chronic stress"and "prolonged exposure to chronic stress" are redundant, as "chronic" already implies a long duration. Consider using either "chronic stress" or "prolonged exposure to stress."
- We have replaced"long-term chronic stress"and "prolonged exposure to chronic stress" with "chronic stress" throughout the manuscript on lines 37, 47, and 90.
- The structure of the phrase "may lead to the development of stress-related insomnia into chronic insomnia"is confusing. Suggested alternatives include:
"…may lead to the progression of stress-related insomnia into chronic insomnia"
"…may cause stress-related insomnia to become chronic insomnia"
- We have replaced"may lead to the development of stress-related insomnia into chronic insomnia" with "…may lead to the progression of stress-related insomnia into chronic insomnia" in the texton lines 41 and 42.
- The phrase "Sleep disorders of this nature are typically characterized by…"is somewhat ambiguous. Replace with: "These sleep disturbances are typically characterized by…"
- We have replaced"Sleep disorders of this nature are typically characterized by…" with "These sleep disturbances are typically characterized by…" in the texton line 52.
- Correct the typo "teh"to ""
- We have corrected"teh"with "the" throughout the manuscript.
- Replace "and so on"with a more formal alternative: "among others."
- We have replaced"and so on"with "among others."
- The phrase "the preparation method utilizing the combined CUMS/CRS model…"is confusing. A clearer alternative would be: "The combined CUMS/CRS model with sleep deprivation is considered to more accurately replicate clinical symptoms of stress-induced insomnia."
- We have replaced"the preparation method utilizing the combined CUMS/CRS model…" with "The combined CUMS/CRS model with sleep deprivation is considered to replicate clinical symptoms of stress-induced insomnia more accurately" in the texton lines 67-69.
- Replace "behaviour texts"with: "behavioral tests" or "behavioral assessments."
- We have replaced"behaviour texts"with: "behavioral tests" throughout the manuscript on lines 87, 133 and 533.
- The phrase "sleep-wake scoring and recordings data"could be more accurately stated as:"sleep-wake cycle scoring and electrophysiological recordings" or "sleep architecture data."
- We have replaced"sleep-wake scoring and recordings data" with "sleep-wake cycle scoring and electrophysiological recordings" in the texton line 88.
- Lastly, consider replacing "mice from the model group"with the more concise: "model mice."
- We have replaced"mice from the model group"with "model mice." throughout the manuscript on lines 89, 124 and 351.
Materials and Methods
Animals:
- The phrase "6–8 week SPF male C57BL/6J mice"should be revised for clarity and completeness. Suggested replacement: "Specific pathogen-free (SPF) male C57BL/6J mice, aged 6–8 weeks and weighing 20 ± 2 g..."
- We have replaced"6–8 week SPF male C57BL/6J mice" with "Specific pathogen-free (SPF) male C57BL/6J mice, aged 6–8 weeks and weighing 20 ± 2 g..." in the texton line 94.
- Add when the animals’ body weight were measured.
The analysis of body weight variation over time presents methodological inconsistencies. First, the results include periods during which food deprivation was applied as a stressor, which could confound the interpretation of body weight changes. Proper assessment of this parameter should include body weight measurements taken at the endpoint, after the stress protocol has been completed. Additionally, body weight measurements collected over time should be analyzed using repeated-measures ANOVA, which allows for accurate evaluation of time-dependent effects and potential group-by-time interactions. Only after confirming a significant interaction or a main effect of the experimental condition can it be concluded that the model influences body weight.
Furthermore, food intake should be monitored throughout the experimental period to determine whether the model affects feeding behavior or whether other physiological mechanisms are involved—especially considering that the food deprivation stressor could have a direct impact on this parameter.
- We apologize for overlooking the many factors that affect weight due to our lack of careful consideration. Based on your comments, we believe that "body weight measurements taken at the endpoint" makes more sense. Therefore, we have added "The mice were weighed on day 29 of the official experiment, after the stress protocol has been completed." in the textonlines 101-102. Body weight data were retained only for day 29 (lines 310-312). Figures have also been replaced in text on line 333.
- In lists of three or more terms, commas should be used consistently for clarity. Replace:"Control, CUMS model and CUMS+Noise model groups"with: "Control, CUMS model, and CUMS+Noise model groups."
- We have replaced "Control, CUMS model and CUMS+Noise model groups"with: "Control, CUMS model, and CUMS+Noise model groups." in text on line 97.
- Replace time notations "ZT0:00–12:00"and "ZT12:00–24:00" with the more concise and standard format: "ZT0–12" and "ZT12–24."
- We have replaced"ZT0:00–12:00"and "ZT12:00–24:00" with "ZT0–12" and "ZT12–24." throughout the manuscript on lines 24, 108, 109, 134, 156, 170, 180, 193, and 551.
Instruments and Reagents:
- Avoid repetition of provider names across the equipment list unless necessary for clarity. Group items from the same manufacturer together when possible.
Group similar equipment by category, such as behavioral equipment, molecular analysis tools, etc., to enhance readability and organization.
- Revisions have been made in accordance with your comments on lines 110-115.
- Replace "Nuldus"with the correct spelling: ""
- We have replaced "Nuldus"with"Noldus." throughout the manuscripton lines 162, 176, 187 and 200.
- Replace "Forced Swimming Bucket"with more precise terminology, such as: "forced swim test apparatus" or "forced swim test device."
- We have replaced "Forced Swimming Bucket" with"forced swim test apparatus" in text on lines 111-112.
CUMS and CUMS+Noise model:
- Improve the first sentences, as they are currently split in an unnatural way. Consider combining them into a single, well-structured sentence with appropriate punctuation to enhance readability.
- Revisions have been made in accordance with your comments on lines 117-119.
- Replace "behavioral texts"with the correct term: "behavioral tests."
- We have replaced"behaviour texts"with: "behavioral tests" throughout the manuscript on lines 87, 133 and 533.
- Replace "Eyeball blood collection method"with a more accurate and professional expression: "Blood was collected via retro-orbital sampling under anesthesia."
- We have replaced "Eyeball blood collection method" with "Blood was collected via retro-orbital sampling under anesthesia." in text on lines 134-135.
Pentobarbital sodium sleep test (PSST):
- Avoid unnecessary repetitions of the word ""Revise sentences to maintain clarity without overusing the term.
- Revisions have been made in accordance with your comments on lines 146-150.
- Include a brief explanation to clarify terminology: "The righting reflex is a commonly used parameter to assess sleep onset in rodent models."
- The suggested sentence has been incorporated into lines 145-146 as per your recommendation.
Behavioral tests:
- All behavioral tests must include a habituation period to the experimental room to minimize stress and variability in the animals’ While you added this information in the open field test description, it is absent from the other behavioral test descriptions. Please ensure that each behavioral test description includes the habituation procedure.
To ensure methodological consistency and reproducibility, all behavioral tests were preceded by a habituation period in the experimental room. Equipment was cleaned between each animal using 75% ethanol to eliminate olfactory cues. Behavioral sessions were recorded and analyzed using [insert name of software, e.g., EthoVision XT, Noldus Information Technology]
Please indicate the time of day at which the behavioral tests were conducted (e.g., during the light or dark phase, or specific ZT times), as this can significantly affect the outcomes due to the circadian activity patterns of rodents.
- We apologize for the inadvertent omission of essential information. Each behavioral methodology section has been rigorously revised in compliance with your recommendations (lines 151-201).
- "Open field text(OFT)"must be replaced by "Open field test (OFT)"
- We have replaced "text" with "test" throughout the manuscripton lines17, 152, 154 and 315.
- The phrase "The open field box was performed..." is incorrect, the tests are performed, no the boxes. Replace by "The open field test was conducted...".
- We have replaced "The open field box was performed..."with "The open field test was conducted..."in text on line 154.
- Number of animals at the same time: saying that 3 mice (2 model and 1 control) were placed simultaneously in different boxes is not entirely clear. It's better to explain it more precisely. Each animal was placed individually, right?
- We apologize for the imprecision in our original articulation. The content has been revised in compliance with your expert guidance (lines 156-158).
- Considering that the sand in the device can be virtually divided into 3 zones (center, intermediate, and periphery), it would be appropriate to include a figure indicating the dimensions of the central zone.
In the Open Field Test (OFT), the CUMS group showed a statistically significant reduction in both total distance moved and number of crossed lines. These findings suggest decreased locomotor activity but do not necessarily indicate increased anxiety-like behavior. To appropriately assess anxiety, the analysis should be normalized. While you mention mean velocity and time spent in the center, a more rigorous approach would include quantifying the distance traveled, number of entries, and time spent in three virtual zones (central, intermediate, and peripheral). These parameters should then be normalized to the total distance traveled or total entries, as appropriate.
- We agree with your comments, which can avoid false positives. Based on your comments, we divided the open-field box into three zones (center, intermediate, and periphery). The total distance traveled and the crossing grid count in the center area have been normalizedin text on lines 162-163, 315-320.
Elevated plus maze test(EPM):
- Redundancy: The description of the EPM structure is repeated twice.
- The redundant sentence has been removed to enhance methodological clarity (lines 164-167).
- It does not mention the height of the closed walls.
- The height of the closed wallshave been added in text on line 167.
- "At the beginning of the text" must be replaced by "At the beginning of the test."
- We have replaced "At the beginning of the text"with "At the beginning of the test."in text on line 170.
- Replace "Central open area" by "center" or "central area."
- We have replaced "Central open area"with "central area"in text on line 171.
- The description of the "preference for open arms" must be improved. "Preference for being in open arms..." should be replaced by "percentage of entries into open arms and time spent there."
The time spent in the open arms is presented as one of the evaluated parameters; however, this was not mentioned in the Methods section. There, it is stated that "the preference for being in open arms over closed arms (expressed as a percentage of entries to open arms) was analyzed." This description must be expanded to include both the number of entries and the time spent in the open arms, as both are standard and complementary measures of anxiety-like behavior in the elevated plus maze.
- We have replaced "preference for open arms "with "percentage of entries into open arms and time spent"in text on lines 175-176.
Tail suspension test(TST)
- "had their tails suspended by a restraint band located 2 cm from the tail's end" should be replaced by "were suspended by the tail using adhesive tape placed 2 cm from the tip."
- We have replaced "had their tails suspended by a restraint band located 2 cm from the tail's end" with "were suspended by the tail using adhesive tape placed 2 cm from the tip." in text on lines 179-181.
- "individuals were placed..." should be replaced by "mice were placed..."
- We have replaced "individuals were placed..." with "mice were placed..." in text on lines 194.
- Eliminate the unnecessary repetition of "immobility."
- Revisions have been made in accordance with your comments on lines 185-190.
- "delineates the duration..." sounds confusing and should be improved. Consider replacing it with "defines the duration..." or "indicates the duration..." depending on the context.
- We have replaced "delineates the duration..." with "defines the duration..." in text on line 187.
Forced swimming test(FST):
- Redundancy in the definition of immobility: it is explained twice with similar phrases. You can remove the repetition and consolidate the explanation in one clear sentence.
- The redundant sentence has been removed in accordance with your comments on lines 199-200.
- "Passive suspension" doesn't fit here (it applies well in TST). It should be replaced by "minimal movements necessary to keep the head above water."
- We have replaced "Passive suspension"with "minimal movements necessary to keep the head above water."in text on lines 199-200.
- Replace "Positioned within a Plexiglas cylinder" with "placed in a Plexiglas cylinder."
- We have replaced "Positioned within a Plexiglas cylinder" with "placed in a Plexiglas cylinder."in text on line 195.
- Add water height: if available, ideally around 15 cm, to prevent the mice from touching the bottom.
- The water height have been added in accordance with your comments on lines 196.
- Replace "analysed" with "analyzed."
- We have replaced "analysed" with "analyzed." throughout the manuscripton lines176, 186, 200 and 252.
- Logical order: first describe the test, then the analysis.
- Revisions have been made in accordance with your comments on lines 191-201.
Sleep recordings and scoring:
- Confusing or contradictory phrase: "Automatic scoring > 0 is sleep and automatic scoring < 0 is sleep." This sentence is incorrect as both conditions cannot represent "sleep." One should likely refer to "wake" and the other to "sleep." The phrase should be revised to: "Automatic scoring > 0 represents wake, and automatic scoring < 0 represents sleep," assuming that’s the intended logic. If this is not the case, you might need to check for a typographical error.
- We apologize for the dual mislabeling of "Wake" as 'Sleep' in the text. Revisions have been made on lines 208-211.
- Incorrect or exaggerated statement: "Piezoelectric systems have been validated to be up to 90% more accurate than EEG/EMG-based sleep recording." This is incorrect. Piezoelectric systems have been validated to show up to ~90% concordance with EEG/EMG, but they are not more accurate. The statement should be modified to: "Piezoelectric systems have been validated to show up to ~90% concordance with EEG/EMG-based sleep recordings, but EEG/EMG remains the gold standard."
- We agree with your comments and have replaced it with "Piezoelectric systems have been validated to show up to ~90% concordance with EEG/EMG-based sleep recordings, but EEG/EMG remains the gold standard."on lines 208-211.
- Non-standard or confusing terms: The terms "array duration" and "arousal array duration" are not commonly used in this context. It is better to clarify that you are referring to the duration of wake or sleep episodes. For example: "The duration of wake episodes" or "The duration of sleep episodes." The term "average array duration" is ambiguous and should be rephrased as "the average duration of wake or sleep episodes."
- We have replaced "array duration"with "The duration of wake episodes" or "The duration of sleep episodes."in text on lines 215-217.
- "The detection time is the 12h dark phase, ZT12:00–ZT24:00, i.e. 20:00 to 8:00." There is a time discrepancy. ZT12:00 to ZT24:00 is from 20:00 to 8:00 only in an inverted cycle, which hasn’t been clarified. The time description should be revised for clarity: "The detection time occurs during the 12-hour dark phase, ZT12:00–ZT24:00, corresponding to 20:00 to 8:00 in a standard cycle where ZT0 = lights on at 8:00 and ZT12 = lights off at 20:00." Furthermore, if sleep detection was performed during the dark phase, this is a mistake because mice are nocturnal animals.
- We have replaced "The detection time is the 12h dark phase, ZT12:00–ZT24:00, i.e. 20:00 to 8:00." with"The detection time occurs during the 12-hour dark phase, ZT12:00–ZT24:00, corresponding to 20:00 to 8:00 in a standard cycle where ZT0 = lights on at 8:00 and ZT12 = lights off at 20:00."on lines 217-219.
- As it was not possible to maintain a quieter environment during the light phase, this could result in errors in the data obtained. Therefore, we decided to monitor only the sleep status of the mice in the dark phase. As we were not able to perform 24 h sleep monitoring, the experimental results are limited. However, we still found significant differences in sleep duration and circadian rhythms between themodelmice and the control group, even in the dark phase. The sleep disturbances observed in the model mice were characterized by shorter sleep duration, more fragmented sleep, and sleep rebound. This is consistent with previous findings in models of stress-induced insomnia [1].
[1] Yang Y, Tiliwaerde M, Gao N, et al. Mechanism of GW117 antidepressant action: Melatonin receptor-mediated regulation of sleep rhythm[J]. European Journal of Pharmacology, 2024, 964: 176299.
RNA sequencing:
- Confusing phrase: "specific chemicals to ensure that only RNA was used as template." This is vague. A clearer way to describe the process would be: "Random hexamer primers and reverse transcriptase were used to synthesize cDNA from RNA, and inhibitors of DNA contamination were included to prevent the use of DNA as a template."
- We have replaced "specific chemicals to ensure that only RNA was used as template."with "Random hexamer primers and reverse transcriptase were used to synthesize cDNA from RNA, and inhibitors of DNA contamination were included to prevent the use of DNA as a template."in text on lines 225-227.
- Clarification on estimated size: "The estimated size is 250–300 bp." While this is likely referring to the size of the final fragment of the library, it should be clarified further: "The estimated size of the final PCR product is between 250–300 bp."
- We have replaced "The estimated size is 250–300 bp."with "The estimated size of the final PCR product is between 250–300 bp."in text on line 227.
- Redundant phrase: "The PCR product is purified again..." The word "again" is redundant if it hasn't been mentioned before. It can be rewritten as: "The PCR product is purified," or, if this is the second purification, "The PCR product is purified a second time..."
- The redundantword "again"has been removed. We have replaced "The PCR product is purified again..." with "The PCR product was purified," on line 228.
- Informal wording: "the libraries have been checked and approved." This sounds informal and lacks detail. A more formal and accurate description would be: "The quality and quantity of the libraries were assessed using Bioanalyzer or qPCR, and subsequently, the libraries were pooled."
- We have replaced "the libraries have been checked and approved." with "The quality and quantity of the libraries were assessed using Bioanalyzer or qPCR, and subsequently, the libraries were pooled." in text on lines 229-230.
WGCNA analysis and key module bioinformatics analysis:
- First-person plural: To avoid the loss of fluency caused by the first-person plural, revise the text to maintain a more neutral tone. For example, change: "We conducted the analysis..." to "The analysis was conducted..."
- We agree with the your comments that excessive use of the first-person pronoun diminished objectivity. The text has been systematically revised to ensure academic neutrality and fluency throughout the manuscript.
- "Interactions between genes and disease-differentiating phenotypes": Replace with: "Gene modules associated with phenotypic traits related to disease," which is clearer and more technical, especially in the context of WGCNA (Weighted Gene Co-expression Network Analysis).
- We have replaced "Interactions between genes and disease-differentiating phenotypes"with "genes within the modules showing a significant correlation with phenotypic traits (P < 0.05) were considered to be biologically relevant,"in text on lines 244-246.
- "Disease-differentiating phenotypes": This term is ambiguous. You can improve it by specifying the context more clearly. A better version could be: "Disease-associated phenotypic traits" or "Phenotypes that differentiate disease states."
- We have removed "Disease-differentiating phenotypes".
- "Modules are then clustered and similar modules are combined": This can be made more precise by changing it to: "Modules with similar expression profiles were merged."
- We have replaced "Modules are then clustered and similar modules are combined"with "Modules with similar expression profiles were merged."in text on lines 242-243.
- "P < 0.05 indicated that the genes in the module were consistent with the sample traits": A clearer and more scientifically appropriate version would be: "Genes within modules showing a significant correlation with phenotypic traits (P < 0.05) were considered to be biologically relevant."
- We have replaced "P < 0.05 indicated that the genes in the module were consistent with the sample traits"with "Genes within modules showing a significant correlation with phenotypic traits (P < 0.05) were considered to be biologically relevant."in text on lines 244-246.
- "Heterogeneous expression of candidate genes in both groups was analysed": This phrase is vague. Replace it with: "To identify differentially expressed genes (DEGs), we used..." This ensures clarity and precision.
- We have replaced "Heterogeneous expression of candidate genes in both groups was analysed"with "To identify differentially expressed genes (DEGs), we used..."in text on line 247.
- "DEGs": Include the term "DEGs" after introducing "differentially expressed genes" to maintain consistency. For example: "To identify differentially expressed genes (DEGs), we used..."
- Revisions have been made in accordance with your comments on line 247.
- "Key module genes were then analysed for GO and KEGG…": To improve fluency and avoid redundancy, replace with: "Genes from key modules were further analyzed for GO and KEGG..."
- We have replaced "Key module genes were then analysed for GO and KEGG…"with "Genes from key modules were further analyzed for GO and KEGG..."on lines 251-252.
- "stringApp (v2.1.1) plug-in": Use the more standard phrasing: "stringApp plugin (v2.1.1)."
- We have replaced "stringApp (v2.1.1) plug-in"with "stringApp plugin (v2.1.1)."on line 254.
Nissl staining:
- Atlas and Coordinates: Include which anatomical atlas was used for the brain sections. For example: "Brain sections were performed according to the [specific atlas name] and coordinates relative to bregma were used (e.g., AP: +1.5 mm, or as appropriate for the region of interest)."
- "The brain sections of the hippocampus were examined, and coordinates relative to bregma were used (AP: -1.3 to -2.9 mm)"have been made in accordance with your comments on lines 267-269.
- To follow a more standard scientific notation, replace "approximately"with the shorthand:"…~5 μ."
- We have replaced "approximately"with "~5 μm"on line 260.
- "Gradient deparaffinisation was followed by Nissl staining"replace with: "Sections were deparaffinized through a graded ethanol series and stained with Nissl stain."
- We have replaced "Gradient deparaffinisation was followed by Nissl staining" with "Sections were deparaffinized through a graded ethanol series and stained with Nissl stain." in text on lines 260-261.
- "Remaining sections": Clarify the meaning of "remaining sections"to avoid confusion. You can revise: "The sections were then treated…" if you are referring to all sections. If referring to specific ones, consider specifying what happened to others (e.g., "The sections that were not used for staining were stored for further analysis").
- We have replaced "Remaining sections" with "The sections were then treated…" on line 262.
- "Differentiation solution": Be more specific about the composition of the differentiation solution. For example: "Differentiation was carried out using an acid alcohol solution (e.g., 70% ethanol with 0.1% acetic acid) to remove excess stain."
- We have added more specific about the composition of the differentiation solutionin text on lines 262-264.
- "Sealed with neutral adhesive"Replace with: "Mounted with neutral resin" (e.g., Permount or similar). This is more specific and standard in histology.
- We have replaced "Sealed with neutral adhesive" with "Mounted with neutral resin" on lines 264-265.
- "Neutral adhesive": As mentioned, replace with "neutral resin" or specify the brand if known (e.g., Permount, DPX).
- We have replaced "Neutral adhesive" with "neutral resin" on line 265.
- "Finally, microscopic examination was performed and images were captured and analysed": This can be made more precise by revising to: "Finally, stained sections were examined under a light microscope, and representative images were captured and analyzed."
- We have reversed "Finally, microscopic examination was performed and images were captured and analysed" to "Finally, stained sections were examined under a light microscope, and representative images were captured and analyzed." on lines 265-266.
- Microscopic Examination Details: If possible, add further detail about the microscope used, such as: "Microscopic examination was performed using a [type of microscope] with [magnification, e.g., 10x, 40x] magnification."
- Revisions have been made in accordance with your comments on lines 268-269.
RNA extraction and quantitative RT-PCR detection:
- "authenticity of the results" replace with: "reliability," "accuracy," or "validity" depending on the specific context. For example, "We aimed to assess the reliability of the results."
- We have replaced "authenticity of the results"with"reliability," on line 284.
- "high-throughput detection": Replace with: "high-throughput sequencing" to properly describe RNA-seq or similar methods.
- We have replaced "high-throughput detection"with "high-throughput sequencing"on 284.
- "the prediction of mRNA expression by bioinformatics analysis": Replace with: "the prediction of gene expression profiles by bioinformatics analyses" for more precision and correct plural usage.
- We have replaced "the prediction of mRNA expression by bioinformatics analysis"with "the prediction of gene expression profiles by bioinformatics analyses"on lines 285.
- "verify differentially 3 expressed mRNAs": Replace with: "validate three differentially expressed mRNAs" to correct the grammatical error and use the appropriate verb in this context.
- We have replaced "verify differentially 3 expressed mRNAs"with "validate three differentially expressed genes"on line 286.
- "including the up-regulated mRNA in the CUMS+Noise group: Alb, and the down-regulated mRNAs in the CUMS+Noise group: Npsr1 and P2rx1." Replace with: "including the upregulated gene Alb, and the downregulated genes Npsr1 and P2rx1, all in the CUMS+Noise group." This is more precise as it refers to genes, not mRNA, and avoids repetition of "CUMS+Noise group."
- We have replaced "including the up-regulated mRNA in the CUMS+Noise group: Alb, and the down-regulated mRNAs in the CUMS+Noise group: Npsr1 and P2rx1." with "including the upregulated gene Alb, and the downregulated genes Npsr1and P2rx1, all in the CUMS+Noise group." on lines 286-288.
- "cDNA was amplified using...": Modify as: "cDNA was synthesized using [kit name], followed by amplification with qPCR." This distinction clarifies that amplification occurs during the qPCR step, not during cDNA synthesis.
- Revisions have been made in accordance with your comments on lines 293-294.
- "ABI 7300 real-time qPCR system": Replace with: "ABI 7300 Real-Time PCR System" for correct capitalization.
- We have replaced "ABI 7300 real-time qPCR system"with "ABI 7300 Real-Time PCR System"on line 295.
- "comparative CT method (2-ΔΔCT)": Replace with: "comparative Ct method (2^–ΔΔCt)" to correct the notation for the method used in qPCR analysis.
- We have replaced "comparative CT method (2-ΔΔCT)"with "comparative Ct method (2^–ΔΔCt)"on line 296.
- "and t-test was used to evaluate the results":Replace with: "and statistical significance was assessed using a Student’s t-test" for clarity and proper scientific phrasing.
- We have replaced "and t-test was used to evaluate the results"with "and statistical significance was assessed using a Student’s t-test"on line 298.
Statistical analysis
- "mean±standard deviation": Correct format: "mean ± standard deviation" with spaces before and after the "±" symbol, as per typographic conventions. Also, SD abbreviation is also used with sleep deprivation.
- We have replaced "mean±standard deviation"with "mean ± standard deviation". SD abbreviation is onlyused with sleep deprivation. Revisions are in text on lines 300, 344 and 498.
- "Differences between groups were determined by...": Replace with: "Differences between groups were assessed using..." This is more common and technically correct in this context.
- We have replaced "Differences between groups were determined by..." with "Differences between groups were assessed using..." on lines 301-302.
- "Kruskal Wallis": Corrected to: "Kruskal–Wallis" (with an en dash between "Kruskal" and "Wallis").
- We have replaced "Kruskal Wallis" with "Kruskal–Wallis" on line 301-302.
- "Wilcoxon tests": Specify the type of test as: "Wilcoxon rank-sum tests" to indicate that this is the non-parametric equivalent of the t-test.
- We have replaced "Wilcoxon tests"with "Wilcoxon rank-sum tests"on line 303.
- "relationship": Replace with: "association" or "correlation," as "relationship" is too informal in scientific writing.
- We have replaced "relationship" with "association" or "correlation" throughout the manuscript on lines 466, 577 and 605.
- "key modular hub genes": Replace with: "key module hub genes" for accuracy, as "modular" is incorrect in the context of WGCNA. The term "module hub genes" is the correct terminology.
- We have replaced "modular"with "module"throughout the manuscript on lines 236 and 305.
- "signifcance": Correct the typo: "significance."
- We have corrected "signifcance"to "significance."throughout the manuscript on lines 297 and 306.
Results
- In general, the values of mean ± SD are included both in the main text and in the supplementary tables. This redundancy makes the manuscript difficult to read. You should indicate only the statistical differences in the main text and refer to the corresponding figures and supplementary tables for detailed values.
- We agree your comments and have removed “mean ± SD”in the main textthroughout the manuscript on lines 310-332, 349-383, 395-413 and 480-492.
- In the Elevated Plus Maze (EPM), total distance traveled must also be reported. If differences in this parameter are observed, all other behavioral measures (e.g., entries and time spent in open/closed arms) should be normalized accordingly. Consider including the distance traveled specifically in the open arms.
- We agree with your comments and have analyzed the total distance traveled. There were no significant differences in the data. Details have been placed in the Table S3.
- Additionally, the abbreviations used for arm-related parameters are unclear. It is recommended to adopt standard terminology such as OAE (Open Arm Entries), OAT (Open Arm Time), and OAD (Open Arm Distance).
Regarding Figure 1, the legend should be revised to include the standard abbreviations (e.g., OAT for open arm time, OAE for open arm entries), which facilitates clarity and consistency with the rest of the manuscript.
- We have replaced "OE%" and "OT%" with "OAE%" and "OAT%"throughout the manuscript on lines 324, 333, 404, 411, 472, 491, 567 and 637.
- In addition, the methodology for corticosterone determination is entirely missing from the Methods section. It is essential to indicate how corticosterone was measured (e.g., ELISA, RIA), whether animals were fasted prior to sampling, and at what time of day euthanasia was performed, as these factors can significantly influence hormone levels.
- The methodology for Serum corticosterone measurement have added in the text on lines 270-282. "whether animals were fasted prior to sampling, and at what time of day euthanasia was performed" have added in the text on lines 133-135.
- On Sleep disturbances successfully induced by CUMS+ Noise 8h method:
-It is unclear whether the measurements presented in Figure 2C are repeated measures. If so, the data should be analyzed using a repeated measures ANOVA, as this statistical test is appropriate for accounting for within-subject variability over time or conditions.
- These data are not repeated measure. One mouse was measured once for 12h.
- -Use "..exhibited by the mice"instead of "displayed by the mice"
-Replace "pentobarbital sodium sleep test" by "pentobarbital sodium-induced sleep test".
-Replace "markedly increased" by "significantly increased"
-Replace "notable statistical differences" by "significant differences"
-Replace "computerised" by "computerized"
-Replace "percentage of dark phase sleep" by "percentage of sleep during the dark phase"
-Replace "was found to be significantly reduced" by "was significantly reduced"
- We have replaced the words that were inaccurately described throughout the manuscript. They are in text on lines 349, 351, 353, 354, 355 and 358.
- The terms "down-regulated"and "up-regulated" are generally used in the context of gene expression and are not appropriate for describing changes in behavioral parameters such as sleep episode duration. These should be replaced by "decreased" and "increased," respectively, to accurately reflect the nature of the observed changes in behavior.
- We have replaced "down-regulated"and "up-regulated" with "decreased"and "increased," on lines 362, 364 and 368.
- -Replace "stresses"by "stressors"
-Replace "showed" by "shown"
- We have replaced "stresses"with"stressors" on line 371 and replaced "showed"by "shown" on line 372.
- Figure 2 legend: which is the microscope magnification used in D images?
- We have added "the microscope magnification" in text on lines 269 and 390.
On Key gene module identification:
- Replace "After pretreatment"by "After preprocessing". In bioinformatics, "preprocessing" is a more appropriate term than ""
- We have replaced "After pretreatment"by "After preprocessing" on line 396.
- Replace "variables"by "phenotypes" / "traits"
- We have replaced "variables"by "phenotypes" on line 408.
- Replace "TPM data... was analyzed and log2 was calculated"by "TPM data were logâ‚‚-transformed and used..."
This adjustment improves the accuracy, as it refers to a transformation, not a simple calculation, and ensures proper concordance ("data" is plural in scientific English).
- We have replaced "TPM data... was analyzed and log2 was calculated"by "TPM data were logâ‚‚-transformed and used..." on line 397.
On Key gene module expression and gene function:
- Remove "obviously". Removing "obviously" makes the tone more formal and appropriate for scientific writing.
- We have removed "obviously" throughout the manuscript.
- Replace "DOTPLOT software packages"by "dotplot functions in RStudio". This adjustment clarifies that it's a function from the clusterProfiler package, not a separate software.
- We have replaced "DOTPLOT software packages"with"dotplot functions in RStudio" on line 431.
- Consistency between analysis names and GO categories (BP, MF, CC).
- Revisions have been made in accordance with your comments on line 433.
On Selection and verification of key module hub genes
- Consistency in gene and pathway names: Genes (e.g., Alb, P2rx1, etc.) should be italicized, and official names for pathways should be used.
- Revisions have been made in accordance with your comments throughout the manuscript.
- Grammar accuracy: For example, "The findings is shown"should be corrected to "The resulting 15 genes… are shown".
- We have corrected"The findings is shown" to "The resulting 15 genes… are shown" on line 458.
- Logical clarity: rearrange the paragraphs to improve the natural progression: selection → functions → correlations → conclusions.
- We have rearrange the paragraphsin accordance with your comments on lines 453-477.
- Academic style: remove repetitions and improved connectors.
- We have removed the repetitionsand improved connectorsin order to read fluently on lines 453-477.
On Validation of the altered expression of hippocampal genes in mice subjected to CUMS in conjunction with noise exposure
- Consistency in style: Gene names should be italicized, pathways in lowercase without quotation marks, and "GO" and "KEGG" should be used as proper nouns.
- We have standardized our writing style on lines 453-477.
- Elimination of repetitions: For example, "The gene was significantly enriched…"repeated twice consecutively.
- We have removed repetitions"The gene was significantly enriched…" in text on lines 479-491.
- More precise conclusion: add "molecular mechanisms underlying…"to reinforce the functional impact of the finding.
- We have added more precise conclusion "molecular mechanisms underlying…" on lines 489-490.
Discussion
- (Line 481–482). sleep-onset insomnia, short-term sleep insomnia. " Short-term sleep insomnia" sounds redundant or unclear. It is recommended to use: "transient insomnia" or simply "short-term insomnia."
- We have replaced " Short-term sleep insomnia"with "short-term insomnia."on line 519.
- (Line 483–486). These also serve as the physiological foundation..." You correctly identified the subject-verb agreement issue — "These" is plural, so the verb must be "serve" instead of "serves".
- We have corrected "serves"to "serve"on line 523.
- (Line 491–492). The specific details of model preparation and regulatory mechanisms are not fully understood.
- We have added "The specific details of model preparation and regulatory mechanisms are not fully understood." on lines 529-530.
- (Line 497). "..manifestations of anxiety-like mood disturbances". Slightly redundant (mood disturbances and anxiety-like).
- We have removed the redundant"mood disturbances" on lines 517-614.
- (Line 511–514). "interval 31–32h and at time interval 1920s"Confusing: Is "1920s" a typo? Maybe it should be "interval 19–20 h" or "1920 s (seconds)"
- We apologize for the ambiguity in the content due to use "interval 31–32h and at time interval 1920s". It has been replaced "at the final time interval of ZT23-24 and at the 1920 s time interval (percentage of single sleep duration)" on lines 551-552.
Conclusión
- Errores detectados y sugerencias de corrección:
Lexical redundancy: "sleep disturbances were specifically characterized by sleep duration and sleep rhythm disturbances". "Sleep disturbances" is repeated twice in the same sentence, creating a feeling of unnecessary repetition.
- We have removed the redundant"Sleep disturbances" on lines 609-614.
Supplementary material
- Add units in the tittle tables
- Add F values in the table description followed of p values. Remove them from the tables.
- There are several mistakes like "text"instead of "test" and typing error.
- Do not repite the ANOVA description every time you mention it.
- Revisions have been made in accordance with your comments on Supplementary Materials: Appendix A.
Comments on the Quality of English Language
- The manuscript addresses a relevant topic; however, I strongly recommend a thorough revision of the English writing. There is an inconsistent use of passive and active voice, variations between first-person singular and plural expressions, and several grammatical errors throughout the text. Additionally, the manuscript shows a mixture of British and American English, which further affects the overall clarity and consistency.
- Weapologize for any inconvenience caused by thepoor English writing. To further improve the readability of the manuscript and make the wording more precise, we had used the professional English editing service recommended by the journal.
We have revised and improved the manuscript in response to the reviewers’ comments. These changes do not affect the content and scope of the study. We have marked the revisions in red in the resubmitted manuscript.
We express our sincere gratitude to the reviewers for their insight comments and suggestions, and we hope that the revisions will meet with approval.
Sincerely yours,
Chang-qing Tong

Reviewer 2 Report
Comments and Suggestions for Authors
Extensive comprehensive research, including a variety of methods and approaches: behavioral, pharmacological, somnological (using effective but rarely used piezoelectric recording), histochemical, molecular biological. The authors have obtained interesting and important results both for fundamental science and, possibly, for the clinic. They identified a possible link between sleep, emotionality, and thyroid hormones. I have no serious complaints, it's a brilliant piece of work, and I strongly recommend its publication as soon as possible. There are only a couple of stylistic remarks:
line 196. “Automatic scoring > 0 is sleep and automatic scoring < 0 is sleep.” Isn’t it an error?
line 510. Should be NREM instead of RNEM
Author Response
Responses to Reviewers' comments
Dear Reviewers,
We are very grateful for your comments on our manuscript entitled "Hippocampal transcriptome analysis in a mouse model of chronic unpredictable stress insomnia" (Biomedicines-3521469). These comments have been very valuable in helping us revise and improve the manuscript. We sincerely appreciate your positive evaluation of our work. We also have carefully considered these comments and revised the manuscript accordingly. According to the submission guidelines and your suggestions, we have carefully revised the manuscript and highlighted the revised parts in red in the text. The major revisions and responses to the reviewers' comments are listed below.
Responses
- line 196. “Automatic scoring > 0 is sleep and automatic scoring < 0 is sleep.” Isn’t it an error?
- We apologize for the dual mislabeling of "Wake" as "Sleep" in the text. An automatic score of <0 represents wake, while a score of >0 represents sleep. Revisions have been made on lines 208-210.
- line 510. Should be NREM instead of RNEM
- We apologize for using the wrong word “RNEM”due to our oversight. It have replaced with “NREM”in text on line 548.
We have revised and improved the manuscript in response to the reviewers’ comments. These changes do not affect the content and scope of the study. We have marked the revisions in red in the resubmitted manuscript.
We express our sincere gratitude to the reviewers for their insight comments and suggestions, and we hope that the revisions will meet with approval.
Sincerely yours,
Chang-qing Tong

Round 2
Reviewer 1 Report
Comments and Suggestions for Authors
The manuscript has been improved, and almost all of my previous comments have been addressed. However, I still have a few questions and suggestions:
Abstract
Overall, it is acceptable. Lines 17-18 which is the model group?
Lines 27–32 should be revised to improve the flow and clarity.
Introduction
Generally well written, but lines 52–58 should be revised due to the repetition of certain expressions, which generates confusion. The use of contrastive language suggests an opposition when in fact the relationship between mood disorders and insomnia is likely bidirectional or mutually influential.
Please revise the conceptual continuity throughout the section; in some parts, the text lacks a coherent flow.
Recommendation: I suggest explicitly stating the aim of the manuscript. This will help clarify the primary objective and provide readers with a clearer understanding of the study’s purpose.
Methods
Line 93: Please specify the number of animals per group, e.g., "n = … per group". Why does the total n=30 if there were 9 animals per group?
Line 98: The sentence is confusing. Clarify that animals were weighed at the end of the stress protocol (i.e., on day 29).
Lines 106–111: These paragraphs lack a main verb and need grammatical revision.
Lines 116–123: Indicate clearly whether the stressors were administered daily for 28 days. Also, replace “stimuli” with “stressors” for consistency and precision.
Lines 128–129: This sentence should be revised to reduce confusion caused by the multiple time references.
Lines 155–158: You mention that the open field arena was divided, but this is not described elsewhere in the manuscript. Also, I believe my previous comment was misunderstood. First, determine if there are differences in total distance traveled and report this value. Then, normalize the other parameters (e.g., distance in center/periphery) accordingly. I strongly recommend focusing on time and distance spent in the center and periphery rather than velocity, which reflects locomotion but provides limited information about anxiety-like behavior. I believe that including the distance traveled and the time spent in each zone would significantly enhance the relevance of the manuscript.
Lines 161–163: The description of the elevated plus maze is unclear—currently, it suggests that all arms have walls. Please clarify. Also, add information on total distance traveled.
Lines 196–213: The discussion of dark-phase-only monitoring is relevant. Please briefly mention this limitation earlier in the methods section (around lines 211–213).
Lines 250–260: This paragraph repeats “the section” multiple times, which hinders readability. Please revise.
Lines 263–264: Include more details about the ELISA procedure (e.g., incubation time and temperature). Also indicate the standard concentrations used (with units), the sample volumes, and the buffer compositions.
Results
Lines 301–302: The statement that “body weight gradually decreased” is inaccurate unless you monitored weight across multiple time points and used repeated measures ANOVA. Additionally, as mentioned in the previous review, food consumption should be included to contextualize the final body weight, especially since food deprivation was used as a stressor. How many instances of food deprivation were administered to the mice during the experiment?
In the OFT, you found significant differences in total distance. Therefore, normalization of the behavioral parameters is necessary (as described in the methods). However, I would like to reiterate my previous observation regarding the inclusion of additional parameters. Additionally, you could mention that the lack of significant differences in mean velocity suggests that the animals did not exhibit locomotion alterations. In panel D of Figure 1, both OFT and EPM parameters are included, but these should be clearly separated for clarity.
In the EPM, include total distance traveled. The same comment about normalization applies—if a mouse spent more time in the open arms, it should also have covered more distance in that area.
Line 313: Add a reference to Figure 1.
Figure 1H: Why is n=5 per group shown for corticosterone results? If this is a representative subset, please clarify this in the methods section.
Discussion
Line 451: The sentence is redundant and should be rephrased.
Lines 557–559: The results are presented unclearly—please revise for better understanding.
Supplemental material
F-values and degrees of freedom should be reported for all statistical tests.
Comments on the Quality of English Language
While the English has been improved, I recommend a more thorough revision aimed at ensuring a smoother flow throughout the text and avoiding repetitive wording.
Author Response
Responses to Reviewers' comments
Dear Reviewers,
We are very grateful for your comments on our manuscript entitled "Hippocampal transcriptome analysis in a mouse model of chronic unpredictable stress insomnia" (Biomedicines-3521469). These comments have been very valuable in helping us revise and improve the manuscript. We have carefully considered these comments and revised the manuscript accordingly. According to the submission guidelines and your suggestions, we have carefully revised the manuscript and highlighted the revised parts in red in the text. The major revisions and responses to the reviewers' comments are listed below.
Responses
Abstract
- Overall, it is acceptable. Lines 17-18 which is the model group?
- Weapologize for not having clarified this.We have revised it to "The results suggested that the mice in both model groups exhibited anxiety-like behavior." in text on lines 17-18.
- Lines 27–32 should be revised to improve the flow and clarity.
- We apologize for any inconvenience caused by the poor English writing.To further improve the readability of the manuscript and make the wording more precise, we invited a professional English translation professor to help revise this manuscript. We revise it on lines 27-33.
Instruction
- Generally well written, but lines 52–58 should be revised due to the repetition of certain expressions, which generates confusion. The use of contrastive language suggests an opposition when in fact the relationship between mood disorders and insomnia is likely bidirectional or mutually influential.
- We agree with your comments on "the relationship between mood disorders and insomnia is likely bidirectional or mutually influential". We revisedit in the text on lines 52-59.
- Please revise the conceptual continuity throughout the section; in some parts, the text lacks a coherent flow.
- Revisions have been made in accordance with your comments on lines 60-82.
- Recommendation: I suggest explicitly stating the aim of the manuscript. This will help clarify the primary objective and provide readers with a clearer understanding of the study’s purpose.
- We have added "This study aimed to establish a fitting animal model for insomnia triggered by stress. On this basis, underlying mechanisms of interaction between stress-related insomnia and anxiety-like behavior were explored. " in the textonlines 89-92.
Methods
Animals:
- Line 93: Please specify the number of animals per group, e.g., "n = … per group". Why does the total n=30 if there were 9 animals per group?
- We apologize for the death of one mouse in each group due to our mishandling. We have replaced "27"with "30"in the text on line 98.
- Line 98: The sentence is confusing. Clarify that animals were weighed at the end of the stress protocol (i.e., on day 29).
Line 313: Add a reference to Figure 1.
Lines 301–302: The statement that “body weight gradually decreased” is inaccurate unless you monitored weight across multiple time points and used repeated measures ANOVA. Additionally, as mentioned in the previous review, food consumption should be included to contextualize the final body weight, especially since food deprivation was used as a stressor. How many instances of food deprivation were administered to the mice during the experiment?
- We agree with your comments. Because we ignored the effect of fasting on body weight, we did not count the amount of food eaten by the mice. This could lead to an incorrect result. We request that the body weight results be removed to ensure the accuracy of the manuscript data. We hope you will approve it.
- Lines 106–111: These paragraphs lack a main verb and need grammatical revision.
- Revisions have been made in accordance with your comments on lines 103-110.
- Lines 116–123: Indicate clearly whether the stressors were administered daily for 28 days. Also, replace “stimuli” with “stressors” for consistency and precision.
- We have replaced “stimuli” with “stressors” on lines 122, 126, 128 and 130.
- Lines 128–129: This sentence should be revised to reduce confusion caused by the multiple time references.
- Revisions have been made in accordance with your comments on lines 125-130.
- Lines 155–158: You mention that the open field arena was divided, but this is not described elsewhere in the manuscript. Also, I believe my previous comment was misunderstood. First, determine if there are differences in total distance traveled and report this value. Then, normalize the other parameters (e.g., distance in center/periphery) accordingly. I strongly recommend focusing on time and distance spent in the center and periphery rather than velocity, which reflects locomotion but provides limited information about anxiety-like behavior. I believe that including the distance traveled and the time spent in each zone would significantly enhance the relevance of the manuscript.
- I apologize that I did misunderstand your comments. We have replaced "percentage of total distance moved in the center (TMDC%) , time spent in the center (TPC), percentage of total distance moved in the periphery (TMDP%), and time spent in the periphery (TPP)."with "mean velocityand crossing grid count" in the text on lines 162-164.
- Lines 161–163: The description of the elevated plus maze is unclear—currently, it suggests that all arms have walls. Please clarify. Also, add information on total distance traveled.
- Revisions have been made in accordance with your comments on lines 175-177.
- Lines 196–213: The discussion of dark-phase-only monitoring is relevant. Please briefly mention this limitation earlier in the methods section (around lines 211–213).
- We have added "during the 12-hour dark phase"in the text on line 213.
- Lines 250–260: This paragraph repeats “the section” multiple times, which hinders readability. Please revise.
- We have removed repeats “the section”in the text on lines 260-270.
- Lines 263–264: Include more details about the ELISA procedure (e.g., incubation time and temperature). Also indicate the standard concentrations used (with units), the sample volumes, and the buffer compositions.
- Revisions have been made in accordance with your comments on lines 271-286.
Results
- In the OFT, you found significant differences in total distance. Therefore, normalization of the behavioral parameters is necessary (as described in the methods). However, I would like to reiterate my previous observation regarding the inclusion of additional parameters. Additionally, you could mention that the lack of significant differences in mean velocity suggests that the animals did not exhibit locomotion alterations. In panel D of Figure 1, both OFT and EPM parameters are included, but these should be clearly separated for clarity.
- We have replaced "percentage of total distance moved in the center (TMDC%) , time spent in the center (TPC), percentage of total distance moved in the periphery (TMDP%), and time spent in the periphery (TPP)."with "mean velocityand crossing grid count" in the text on lines 175-177, and 314-333. In Figure. 1, we have also made a clear separation between the OFT and EPM parameters on line 334.
- In the EPM, include total distance traveled. The same comment about normalization applies—if a mouse spent more time in the open arms, it should also have covered more distance in that area.
- We apologize again for misunderstanding your comments. We mistakenly assumed that it was necessary to calculate the total distance moved by all the arms, and therefore found no difference. Revisions have been made in accordance with your comments on lines 175-177, 321-326, and 334-348.
- Figure 1H: Why is n=5 per group shown for corticosterone results? If this is a representative subset, please clarify this in the methods section.
- We apologize that due to our unskilled blood sampling collection, not all samples passed quality tests. As a result, only five samples remained from each group.
Discussion
- Line 451: The sentence is redundant and should be rephrased.
- We have removed redundantdescriptions in the text on lines 460-466.
- Lines 557–559: The results are presented unclearly—please revise for better understanding.
- We found inconsistencies between the content of the manuscript in the cited lines and the content of the needed revisions. Therefor, we have revised both sections. It was in the text on lines 556-560 and 569-572.
Supplementary material
- F-values and degrees of freedom should be reported for all statistical tests.
- Revisions have been made in accordance with your comments on Supplementary Materials: Appendix A.
We have revised and improved the manuscript in response to the reviewers’ comments. These changes do not affect the content and scope of the study. We have marked the revisions in red in the resubmitted manuscript.
We express our sincere gratitude to the reviewers for their insight comments and suggestions, and we hope that the revisions will meet with approval.
Sincerely yours,
Chang-qing Tong